# MEAN-FIELD CONTINUOUS SEQUENCE PREDICTORS

## ABSTRACT

We propose a novel class of neural differential equation models called *mean-field continuous sequence predictors* (MFPs) for efficiently generating continuous sequences with potentially infinite-order complexity. To address complex inductive biases in time-series data, we employ mean-field dynamics structured through carefully designed graphons. By reframing time-series prediction as mean-field games, we utilize a fictitious play strategy integrated with gradient-descent techniques. This approach exploits the stochastic maximum principle to determine the Nash equilibrium of the system. Both empirical evidence and theoretical analysis underscore the unique advantages of our MFPs, where a collective of continuous predictors achieves highly accurate predictions and consistently outperforms benchmark prior works.

## 1 INTRODUCTION

Modeling spatiotemporal processes provides profound insights into and enhances the ability to predict the behavior of complex systems that evolve across both temporal and spatial dimensions. In recent studies, neural differential equation models (Chen et al., 2019; Tzen & Raginsky, 2019) have demonstrated exceptional generalization capabilities and effectiveness in capturing continuous-time spatiotemporal dynamics, with applications ranging from generative modeling (Song et al., 2021) and quantitative finance (Cohen et al., 2023) to physically-informed neural networks (Iakovlev et al., 2024). Despite their notable performance, existing approaches fail to offer theoretical findings for a key question inherent to continuous time series: *How does the model behave as time granularity becomes finer, ultimately leading to infinite observations?* To answer the question, a viable approach is to directly model data dynamics over continuous intervals with infinite complexity. To this end, this work focuses on employing mean-field games (Lasry & Lions, 2007), to develop an infinite-dimensional predictive decision-making framework, generalizing existing differential equation models.

The mean-field principle, a core philosophy in several scientific fields such as neuroscience (Faugeras et al., 2009), statistical physics (Negele, 1982), and economics (Carmona, 2020; Cardaliaguet & Lehalle, 2018), serves as a powerful tool to model and analyze how large numbers of interacting agents behave strategically in stochastic dynamics, decentralized environments. In the mean-field regime, a continuum of infinitely many agents is expected to satisfy *Nash equilibrium* by individually governing the dynamics of partially observed historical sequential data and collectively interacting with each other to make optimal group decisions for forecasting future events. The central premise of this game-theoretic interpretation of the predictive system can be encapsulated in the following statement: *We extend the continuous-time sequence prediction problem into the formal setting of mean-field games*. In relation to this statement, our contribution is twofold:

- We extend existing differential equation models by proposing mean-field graphon SDEs as a novel framework for modeling sequence predictors. This framework effectively captures the stochastic spatiotemporal dynamics of an infinite continuum of agents, grounded in conjectures from time series analysis (e.g., seasonality). To efficiently solve the mean-field games, we introduce gradient-based FBSDEs, which significantly reduce the computational complexity associated with approximating Nash equilibrium.

- Building on the concentration of empirical measures and the propagation of chaos property, our theoretical analysis clarifies the effect of leakage in past observations on the generalization performance of the mean-field system. We demonstrate that, as the population of agents increases, the coalition produces increasingly accurate and reliable predictions.

Figure 1: **(Left)**. The mean-field predictors are conditioned on a set of labeled past observations $\{\mathbf{u}_n\}_{n \leq N=4} \sim p(\mathrm{u})$. Each spatiotemporal dynamic is interconnected via the neural graphon $\mathbb{W}_{\boldsymbol{\alpha}}$, which leverages inductive biases tailored for continuous sequential data. **(Right)**. In the training, the collective decisions of a coalition of mean-field predictors are calibrated to approximate the target future event interval.

## 2 MEAN-FIELD CONTINUOUS SEQUENCE PREDICTORS

This section introduces a stochastic differential equation model designed to represent a continuous signal of infinite order, incorporating inductive biases in time-series modeling. For simplicity and without loss of generality, **bold-face** notation will be used to omit sub- and superscripts of mathematical objects when appropriate.

**Definition 2.1.** *(Mean-field Graphon SDEs) For the Markovian feedback controls $\boldsymbol{\alpha} : \mathcal{T} \times \mathbb{R}^d \times \Theta \to \mathbb{R}^d$ (i.e., $\boldsymbol{\alpha} \coloneqq \alpha(t, \mathrm{x}; \theta)$) and continuous labels $\mathrm{v} \sim p(\mathrm{u})$, we propose the $\mathbb{R}^d$-valued controlled stochastic differential equations called a mean-field graphon dynamics defined as follows:*

$$d\mathbf{X}_{\mathrm{u}}^{\boldsymbol{\alpha}}(t) = \langle \mathbb{W}_{\boldsymbol{\alpha}}[\nu_{\mathrm{v}}(t)](\mathrm{u}), \boldsymbol{\psi} \rangle(\mathbf{X}_{\mathrm{u}}^{\boldsymbol{\alpha}}(t), \boldsymbol{\alpha})dt + \boldsymbol{b}(t, \mathbf{X}_{\mathrm{u}}^{\boldsymbol{\alpha}}(t), \boldsymbol{\alpha})dt + \sigma_t dW_t^{\mathrm{u}}, \quad \mathbf{X}_{\mathrm{u}}^{\boldsymbol{\alpha}}(0) \coloneqq \mathrm{y}_{\mathrm{u}}, \quad (1)$$

*where a probability measure $\boldsymbol{\nu} \coloneqq \{\nu_{\mathrm{v}}(t)\}_{(\mathrm{v},t) \in \mathbb{O} \times \mathbb{T}}$ serves as a concise representation of the law of dynamics, and $\mathrm{y}_{\mathrm{u}} \sim p(\mathrm{u}, \mathrm{y})$ denotes a continuous representation of past observations.*

The mean-field dynamics presented in Definition 2.1 involves three terms on the right-hand side, with an emphasis on important notions *mean-field predictors* and *neural graphons*, both critical for comprehensive continuous time-series modeling.

*Mean-field Predictor.* The proposed dynamical system incorporates two types of continuity encoding: *locality* (*i.e.*, $t$) and *labeling* (*i.e.*, u). The state variable $\mathbf{X}_{\mathrm{u}}^{\boldsymbol{\alpha}}(t)$, termed a *continuum of predictors* or *mean-field predictors* (**MFPs**), represents a continuous set of information trajectories, each labeled by $\mathrm{u} \sim p(\mathrm{u})$ and initialized from the past observation, $\mathbf{X}_{\mathrm{u}}^{\boldsymbol{\alpha}}(0) = \mathrm{y}_{\mathrm{u}} \sim p(\mathrm{u}, \mathrm{y})$. For instance, a continuum of predictors for the sequence of infinite i.i.d labels $\mathbf{u}_{\infty} \coloneqq \{u_n \sim p(\mathrm{u}); n \leq N \to \infty\}$ in the mean-field regime $\mathbf{X}_{\mathbf{u}_{\infty}}^{\boldsymbol{\alpha}}(0)$ can be interpreted as being conditioned on the **past observational interval**, *i.e.*, the support of the label distribution $p(u)$, with their future causal effect, producing $\mathbf{X}_{\mathbf{u}_K}^{\boldsymbol{\alpha}}(t)$ at **future event interval** being obtained from the dynamics in Eq (1). This demonstrates that the proposed dynamics is well-suited for handling continuous signals, as it processes both input and output in a continuous manner. In processing continuous signals, the closed Markovian control process $\boldsymbol{\alpha}(\cdot; \theta) \in \mathbb{A}$ parameterized by neural networks $\theta \in \Theta$, referred to as a *neural agent*, governs the trajectory of state $\mathbf{X}_{\mathbf{u}_{\infty}}^{\boldsymbol{\alpha}}(t)$. Fig 1 depicts illustrative examples of how the proposed mean-field predictors are sampled (**left**), propagated (**mid**), and utilized to produce future prediction (**right**).

The overarching goal is then to calibrate the trajectory of predictors by determining the optimal neural agent $\boldsymbol{\alpha}^*$ that best approximates the target interval, *e.g.*, $\mathbb{E}_t[\|\mathbb{E}_{\mathbf{u}_{\infty}} \mathbf{X}_{\mathbf{u}_{\infty}}^{\boldsymbol{\alpha}^*}(t) - \mathbf{y}_t\|_E^2] \approx 0$, where decision aggregation $w : \mathcal{O} \to [0, 1]$ captures the collective behavior of mean-field predictors. Section 3 will present a systematic algorithm to fulfill this objective.

*Neural Graphon.* It is widely recognized in the literature that fundamental assumptions of inductive biases, such as *temporal decay*, *cycles*, and *seasonality* are vital for effective time series modeling. To incorporate these into our continuous mean-field system, we introduce a *neural graphon*, a graphon structure parameterized with neural networks, capturing the inherent heterogeneity among predictors.

**Definition 2.2.** *(Neural Graphon) A graphon is a symmetric integrable function defined on $L^2$, $W : \mathcal{O}^2 \to \mathbb{R}$ equipped with $L^2$ norm. For a probability measure $\mu$ defined on $\mathcal{O} \times \mathbb{R}^d$ with bounded second moment, we define a measure-valued function $\mathbb{W}_{\boldsymbol{\alpha}}[\mu](\cdot) : \mathcal{O} \to \mathcal{M}^a$ and a continuous symmetric function $\boldsymbol{\psi}_{\boldsymbol{\alpha}} := \boldsymbol{\psi}(\mathrm{y}, \mathrm{x}, \boldsymbol{\alpha}) := H_{\boldsymbol{\psi}}(\boldsymbol{\alpha}) \mathbf{Proj}_{\mathcal{S}^{d-1}}(\mathrm{y} - \mathrm{x})$ such that the first term in right-hand side of Eq (1) is defined as $\langle \mathbb{W}_{\boldsymbol{\alpha}}[\mu](\mathrm{u}), \boldsymbol{\psi}_{\boldsymbol{\alpha}} \rangle(\mathrm{y}, \boldsymbol{\alpha}) := \mathbb{E}_{\mathrm{v} \sim p(\mathrm{v}), \mathrm{x} \sim \mu}[W_{\boldsymbol{\alpha}}(\mathrm{u}, \mathrm{v}) \boldsymbol{\psi}_{\boldsymbol{\alpha}}(\mathrm{y}, \mathrm{x})] \in \mathbb{R}^d$.*

---

$^a$Please refer to Section 8.1 for the deatils.

For two tuples $(\mathrm{x}, \mathrm{u}) \sim \nu_{\mathrm{u}} \otimes p(u)$ and $(\mathrm{y}, \mathrm{v}) \sim \nu_{\mathrm{v}} \otimes p(v)$, a symmetric function $\psi$ estimates scaled relative dissimilarity between *spatial features* x and y. The neural agent, *i.e.*, $H_{\boldsymbol{\psi}}(\boldsymbol{\alpha})$, then adjusts the importance of dissimilarity by rescaling projected vectors. Meanwhile, the neural graphon $W$ encodes a degree of interaction between temporal variables u and v. Among the various graphon designs available, we propose two structures informed by inductive biases specific to continuous time series. Note

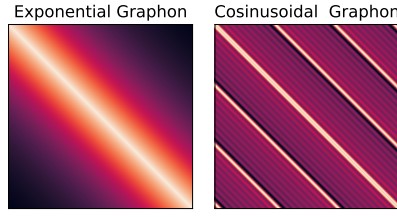

Exponential Graphon    Cosinusoidal Graphon

Figure 2: Visualization of Graphons.

that the key distinction from conventional methods is that our approach *directly models inductive biases in the data space* $\mathbb{R}^d$, rather than in latent feature spaces, facilitated by the graphon structure.

*Exponential Graphon.* In the first graphon structure, we incorporate *temporal decay* (Che et al., 2018) assumption on spatiotemporal variables, which suggests that the influence of the past event decreases exponentially as time deviations increase. Fig 2 shows an illustrative example of the exponential graphon where temporally proximate events tend to exhibit strong interactions, where the neural agent, *i.e.*, $W_1 : \mathbb{A} \to \mathbb{R}^+$ determines the magnitude of interaction. For the deviation between labels $\Delta_{\mathrm{u}} := |\mathrm{u} - \mathrm{v}|$, the impact of temporal dissimilar events are penalized:

$$W_{\boldsymbol{\alpha}}(\mathrm{u}, \mathrm{v}) := W_1(\boldsymbol{\alpha}) \exp(-T^{-1}\Delta_{\mathrm{u}}). \tag{2}$$

*Cosinusoidal Graphon.* The second graphon is designed to emphasize the continuous *cyclic* assumption (Oreshkin et al., 2020), which captures the periodic nature of time-series. To reflect the assumption, we first perform an eigen-decomposition of the proposed graphon operator on $\mathbb{L}^2(\mathcal{O})$, using sinusoidal eigen-functions (*i.e.*, $\{\phi_l\}$) and varying frequency modes for the eigenvalues (*i.e.*, $\{\lambda_l\}$), as Gao & Caines (2019) suggested:

$$\mathbb{W} = \mathbf{Id} + \sum_{k,l \in \mathbb{Z}_+} \lambda_l \varphi_l, \ \{\varphi_l\} \subset \{\mathbf{Id}, \sqrt{2}\cos 2\pi k(\cdot), \ \sqrt{2}\sin 2\pi k(\cdot)\}, \{\lambda_l\} \subset \{a_0, b_k/2\}. \tag{3}$$

We parameterize the graphon operator with neural networks, by replacing Fourier coefficients $\{\mathbf{Id}, \lambda_l\}$ with corresponding neural agents, *i.e.*, $W_0, W_{1,l}, W_{2,l} : \mathbb{A} \to \mathbb{R}^+$. To present various periodicities, we define $\mathfrak{f}(l) \in \{1/2, 1/4, 1/8\}_{l \leq L}$ that represent a set of pre-determined frequencies. We then define *cosinusoidal graphon* as follows:

$$W_{\boldsymbol{\alpha}}(\mathrm{u}, \mathrm{v}) = W_0(\boldsymbol{\alpha}) + \frac{1}{2L} \sum_{l \in \{1, \cdots, L\}} W_{1,l}(\boldsymbol{\alpha}) \cos\left(\frac{2\pi\mathfrak{f}(l)\Delta_{\mathrm{u}}}{|\mathcal{O}|}\right) + W_{2,l}(\boldsymbol{\alpha}) \sin\left(\frac{2\pi\mathfrak{f}(l)\Delta_{\mathrm{u}}}{|\mathcal{O}|}\right). \tag{4}$$

Note that we limit the summation to finite modes (*i.e.*, $L$) for computational tractability. Fig 2 illustrates periodic interaction magnitudes for a predefined frequency setup. Further details on the implementation and their analysis can be found in the Appendix.

## 3 TRAINING MEAN-FIELD PREDICTORS

### 3.1 MEAN-FIELD GAMES AS CONTINUOUS SEQUENCE PREDICTION

In the previous section, we proposed SDE-based mean-field continuous sequence predictors with spatio-temporal interactions. Since the mean-field system in Eq (2.1) is framed as *controlled SDEs* with neural agents, we can formulate the objective function as a *stochastic control problem*. More specifically, our primary goal is to minimize the cost functional $\mathcal{J}$ designed for training neural agents solving sequence prediction and derive the corresponding *value function* $\mathcal{V}$:

**Definition 3.1.** *(Cost functional)[a] For the given neural graphon $\mathbb{W}_{\boldsymbol{\alpha}}$, and fixed set of admissible controls $\mathbb{A}$, the cost functional is defined as follows:*

$$\mathcal{V} := \inf_{\boldsymbol{\alpha} \in \mathbb{A}} \mathcal{J}(\boldsymbol{\nu}^{\boldsymbol{\alpha}}, \boldsymbol{\alpha}) = \inf_{\boldsymbol{\alpha} \in \mathbb{A}} \mathbb{E}_{\boldsymbol{\alpha}, \boldsymbol{\nu}, t \leq T} \left[ \|\mathbb{E}_{u \sim p(u)} \mathbf{X}_u^{\boldsymbol{\alpha}}(t) - y_t\|_E^2 + \mathbf{G}(\mathbf{X}_u^{\boldsymbol{\alpha}}(T), \boldsymbol{\nu}^{\boldsymbol{\alpha}}) \right]. \quad (5)$$

*where $\mathbf{G}$ represents the terminal cost at time $t = T$, and $w : \mathbb{O} \to [0,1]$ is a decision aggregation function, satisfying $\int w(u)du = 1$.*

---

[a]Please refer to Section 8.3 for the detailed rationale of definition.

To generate future predictions, the mean-field predictors collaborate by forming a coalition, *i.e.*, a time marginal of predictors $\mathbb{E}_{u \sim p(u)} \mathbf{X}_u^{\boldsymbol{\alpha}}(t)$, where expectation with respect to labeling u aggregates weighted decisions (*i.e.*, $w$) of a continuum of predictors $u \sim p(u) := w_{\#}[\mathbf{Unif}(\mathbb{O})](u)$ in approximating target continuous interval $\{y_t\}_{t \in \mathbb{T}}$. Figure 1(**right**) provides an illustrative example of the decision-making process. With the aim of generating accurate target intervals, the neural agent is trained to derive *value function $\mathcal{V}$* which characterizes the state in which a continuum of players form a coalition to cooperatively predict the best possible future events.

The challenge in solving this problem stems from the fact that the neural agent both influences the population of predictors $\boldsymbol{\nu}^{\boldsymbol{\alpha}}$, which, in turn, continuously impacts the individual state variables as the dynamics propagate with interactions via neural graphon. To formalize the recurrence in the literature, such problems are often framed as *(graphon) mean-field games* (Lasry & Lions, 2007; Caines & Huang, 2021). In this work, we adopt this approach to formulate **the continuous sequence prediction problem as mean-field games**. Our primary focus is then searching for the best *optimal control $\boldsymbol{\alpha}^*$* that induces the best possible response in the recurrent relation between $\mathcal{V}$ and $\boldsymbol{\nu}^{\boldsymbol{\alpha}}$. For the formal analysis, we investigate how the exact solutions $(\mathcal{V}, \boldsymbol{\nu}^{\boldsymbol{\alpha}^*})$ can be derived from optimal control profiles over time by examining the following system of PDEs in the mean-field regime:

**Definition 3.2.** *(Forward-Backward PDE System). For the obtained optimal neural agent $\boldsymbol{\alpha}^*$, exact solutions of value function in Eq (5) can be obtained by solving the following system of PDEs:*

$$\partial_t \mathcal{V}(t, x) + \sigma_t^2 / 2 \Delta \mathcal{V}(t, x) + H(t, x, \partial_x \mathcal{V}(t, x), \nu_u(t), \boldsymbol{\alpha}^*) = 0, \quad \textbf{(HJB)}$$

$$\partial_t \nu_u^{\boldsymbol{\alpha}^*}(t) - \sigma_t^2 / 2 \Delta \nu_u^{\boldsymbol{\alpha}^*}(t) + \nabla \cdot \left[ \left( \boldsymbol{b}_W(x, \nu_u^{\boldsymbol{\alpha}^*}(t), \boldsymbol{\alpha}^*) + \boldsymbol{b}(t, x, \boldsymbol{\alpha}^*) \right) \nu_u^{\boldsymbol{\alpha}^*}(t) \right] = 0, \quad \textbf{(FPK)}$$

*where $\Delta$ and $\nabla\cdot$ denotes Laplacian and divergence operators, respectively. The stochastic Hamiltonian system $H$ is given by*

$$H(t, x_u, a, \nu, \alpha) := (\boldsymbol{b}_W(x_u, \nu, \alpha) + \boldsymbol{b}(t, x_u, \alpha)) \cdot a + \|\mathbb{E}_{u \sim p(u)} x_u - y_t\|^2, \quad (6)$$

*where $\boldsymbol{b}_W(x, \nu, \alpha) := \langle \mathbb{W}_{\boldsymbol{\alpha}}[\nu](u), \boldsymbol{\psi} \rangle(x, \alpha)$ is the graphon interaction term in Definition 2.2.*

A system of decoupled PDEs consists of the *Hamilton-Jacobi-Bellman* (HJB) equation and the *Fokker-Planck-Kolmogorov* (FPK) equation, which individually describes the propagation rules of the state variable and the value function over time. In mean-field equilibrium states, these PDEs become coupled as the law of the state variables *i.e.*, $\mathbf{Law}(\mathbf{X}_u^{\boldsymbol{\alpha}}(t))$ matches $\nu_u(t)$ with marginal errors. This specific mathematical constraint can be formally expressed in the following definition:

**Definition 3.3.** *(Mean-field $\epsilon$-Equilibrium). We say that a continuous flow of measure $\nu_u(\cdot)$ is an $\epsilon$-equilibrium[a] of graphon mean-field games if there exists a numerical constant $\epsilon > 0$ such that $\sup_{u,t} \left[ \mathcal{W}_2^2(\nu_u(t), \mathbf{Law}(\mathbf{X}_u^{\boldsymbol{\alpha}^*}(t))) \right] \precsim \mathcal{O}(\epsilon)$, such that $\boldsymbol{\alpha}^* \in \mathbb{A}$ is optimal.*

---

[a]Note that the graphon mean-field equilibrium (Zhou et al., 2024) can be recovered by setting $\epsilon = 0$.

The mean-field equilibrium described in Definition 3.3 characterizes a scenario where a continuum of predictors is not incentivized to modify their policies $\boldsymbol{\alpha}^*$ to non-optimal counterpart $\boldsymbol{\beta}$, which induces marginal errors, *i.e.*, $\mathcal{J}(\boldsymbol{\nu}^{\boldsymbol{\beta}}, \boldsymbol{\beta}) \geq \mathcal{J}(\boldsymbol{\nu}^{\boldsymbol{\alpha}^*}, \boldsymbol{\alpha}^*)$. Here, the law of optimal mean-field pre-

Figure 3: Illustrative Algorithm for the Gradient System of FBSDEs.

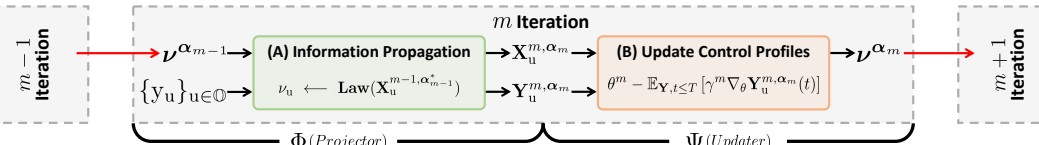

dictors closely approximates the population $\nu_u$ with marginal errors $\epsilon$. This coupling integrates the Hamilton-Jacobi-Bellman (HJB) and Fokker-Planck-Kolmogorov (FPK) equations, forming a *master equation*. Several numerical methods exist to approximate solutions to mean-field games including fixed-point iterations (Lauriere, 2021), and fictitious play (Min & Hu, 2021). However, these methodologies are typically constrained to linear quadratic dynamics, leading to computational intractability when confronting non-linearity (*e.g.*, neural networks). Additionally, numerical simulations for obtaining analytic solutions of this system of PDEs present significant challenges due to the curse of dimensionality in high-dimensional data spaces. The following section is dedicated to addressing these issues by leveraging the deep neural architecture.

## 3.2 Gradient System of Neural Forward-Backward SDEs

Inspired by computational algorithms designed for fictitious play (Cardaliaguet & Hadikhanloo, 2017), we explore a *gradient descent*-based algorithm, which enables us to tackle solving MFGs by fusing deep neural architectures. To be more specific, we propose a gradient system of *forward-backward stochastic differential equations* (Bensoussan et al., 2013), which is adapted for reflecting the update of neural agents with respect to the gradient descent algorithm.

**Definition 3.4.** *(Gradient System of FBSDEs). For the fixed flow of measures $\nu_u(\cdot) : \mathbb{T} \to \mathcal{P}_2$ and the fixed label $u$ at each stage $m$, we consider a family of processes $(\mathbf{X}_u(t), \mathbf{Y}_u(t), \mathbf{Z}_u(t))$ that solves forward-backward stochastic differential equations with respect to the proposed graphon system in Eq* (1) *given as follows:*

$$d\mathbf{X}_u^{m,\boldsymbol{\alpha}_m}(t) = \boldsymbol{b}_W(\mathbf{X}_u^{m,\boldsymbol{\alpha}_m}(t), \nu_u, \boldsymbol{\alpha}_m)dt + \boldsymbol{b}(t, \mathbf{X}_u^{m,\boldsymbol{\alpha}_m}(t), \boldsymbol{\alpha}_m)dt + \sigma_t dW_t^u,$$

$$d\mathbf{Y}_u^{m,\boldsymbol{\alpha}_m}(t) = -H(t, \mathbf{X}_u^{m,\boldsymbol{\alpha}_m}(t), \mathbf{Y}_u^{m,\boldsymbol{\alpha}_m}(t), \nu_u, \boldsymbol{\alpha}_m)dt - \mathbf{Z}_t^m \cdot dW_t^u,$$

$$\boldsymbol{\alpha}_{m+1} := \alpha\left(t, \mathbf{X}_u^{m,\boldsymbol{\alpha}_m}; \theta^m - \mathbb{E}_{\mathbf{Y}, t \leq T}\left[\gamma^m \nabla_\theta \mathbf{Y}_u^{m,\boldsymbol{\alpha}_m}(t)\right]\right) \in \mathbb{A},$$

$$\nu_u = \mathbf{Law}(\mathbf{X}_u^{m-1,\boldsymbol{\alpha}_{m-1}^*}),$$

*where $\gamma^m > 0$ is a learning rate of gradient descent, and $\mathbb{A}$ is a set of admissible neural agents. Then, we have $(\mathbf{Y}_u(t), \mathbf{Y}_u(T), \mathbf{Z}_u(t)) = (\mathcal{J}, \mathbf{G}, (\partial_x \mathcal{J})\sigma_t^{-1})$.*

The proposed gradient system can be decomposed by iterating a two-step procedure, *i.e.*, **(A)** and **(B)**, over a total of $M$ stages. Fig 3 illustrates the evolution of the mean-field predictors related to the updated parameters of neural agents $\boldsymbol{\alpha}_m$ across different stages $m$. The details of the two-step procedure are specified below.

**(A) Information Propagation.** Initially, the system publicly opens the information to a continuum of players by setting the population information of the previous stage, where the forward and backward system of SDEs propagates information with respect to the updated population, $\nu_u$.

$$\nu_u \longleftarrow \mathbf{Law}(\mathbf{X}_u^{m-1,\boldsymbol{\alpha}_{m-1}^*}), \quad (\mathbf{X}_u^m, \mathbf{Y}_u^m) \sim \mathbf{Law}(\mathbf{X}_u^m | \nu_u) \otimes \mathbf{Law}(\mathbf{Y}_u^m | \nu_u). \quad (7)$$

Note that the backward dynamics is propagated in **reverse** direction starting from its terminal state $\mathbf{Y}_u(T) = \mathbf{G}$ while the forward dynamics evolve in the **forward** direction from the initial state. This shows that the proposed FBSDEs parallel the PDE system described in Definition 3.2.

**(B) Update Control Profiles.** In the subsequent step, the neural agent $\boldsymbol{\alpha}^m$ is updated with respect to its parameter $\theta^m$ following the steepest direction of minimizing the values of backward dynamics $\mathbf{Y}_u^m$. The backward dynamics, associated with the cost functional $\mathcal{J}$ as described in Proposition 3.4, guide the updates of the parameters, allowing the mean-field predictors to gradually approximate the

target interval. Since we have proposed an iterative algorithm to solve MFGs, the remaining part aims to provide convergence guarantees and highlight optimality conditions.

**Stochastic Optimality.** Proposition 8.3 guarantees that the gradient system in Definition 3.4 induces optimal neural agents $\boldsymbol{\alpha}^*$, which yield a feasible value function (*i.e.*, $\mathbf{Y}_{\mathrm{u}}^m(0) \xrightarrow{m \to \infty} \mathcal{V}$) where the optimality of the control is represented in the sense of the *Pontryagin stochastic maximum principle* (Yong & Zhou, 2012). Specifically, we have the following two results:

$$\lim_{m \to \infty} \boldsymbol{H}(\,\cdot\,, \boldsymbol{\alpha}_m) \approx \inf_{\boldsymbol{\alpha} \in \mathbb{A}} \boldsymbol{H}(\,\cdot\,, \boldsymbol{\alpha}), \; dt \otimes d\boldsymbol{\nu} - \text{a.e.}, \quad \mathcal{V} \approx \mathbf{Y}_{\mathrm{u}}^{\infty}(0) = \mathcal{J}(\boldsymbol{\nu}^{\boldsymbol{\alpha}_\infty}, \boldsymbol{\alpha}_\infty). \tag{8}$$

The result illuminates that a pair $(\lim_{m \to \infty} \boldsymbol{\alpha}^m = \boldsymbol{\alpha}^*, \lim_{m \to \infty} \boldsymbol{\nu}^{\alpha_m} = \boldsymbol{\nu}^{\boldsymbol{\alpha}^*})$ solves both HJB and FPK equations in Definition 3.2, assuring stochastic optimality. Having obtained the value function, the next goal is to provide an explicit estimation of $\epsilon$ in the convergence of mean-field equilibrium.

**Convergence to Mean-field Equilibrium.** To rigorously analyze the convergence to equilibrium in a distributional sense, we define two distinct operators, $\Phi$ and $\Psi : \mathcal{M} \to \mathcal{M}$, referred to as the *projector* and *updater*, respectively. Each operator corresponds to one of the two steps mentioned earlier, as illustrated in Fig. 3:

$$\Phi(\boldsymbol{\nu}^{\boldsymbol{\alpha}_m}) := \{\mathbf{Law}(\mathbf{X}_{\mathrm{u}}^{\alpha_m}(t))|_{\boldsymbol{\nu} = \boldsymbol{\nu}^{\boldsymbol{\alpha}^*_{m-1}}} \; ; \; t \in \mathbb{T}, \mathrm{u} \in \mathcal{O}\}. \tag{9}$$

$$\Psi(\boldsymbol{\nu}^{\boldsymbol{\alpha}_{m-1}}) := \{\boldsymbol{\nu}^{\boldsymbol{\alpha}_m}; \; \mathcal{V} = \mathcal{J}(\boldsymbol{\nu}^{\boldsymbol{\alpha}^*_{m-1}}, \boldsymbol{\alpha}^*_{m-1}), \; \boldsymbol{\nu}^{\boldsymbol{\alpha}_m} = \boldsymbol{\alpha}^*_{m-1}\}. \tag{10}$$

It can be easily verified that the composition of these operators at stage $m$ maps the previous state's population to the next stage *i.e.*, $\Phi \circ \Psi(\boldsymbol{\nu}^{m-1}) = \boldsymbol{\nu}^m$. Proposition 3.5 asserts that the population $\{\boldsymbol{\nu}^{\boldsymbol{\alpha}_m}\}_{m \leq M}$ generated by the proposed algorithm begins to converge in the Wasserstein metric as the stages $m$ increase.

---

**Proposition 3.5.** *(informal) For arbitrary* $\mathrm{u} \sim p(\mathrm{u})$ *and* $t \in \mathbb{T}$, *the* $m$-*fold of composition* $\Phi \circ \Psi$ *induces convergent behavior of squared 2-Wasserstein distance:*

$$\mathcal{W}_2^2([\Phi \circ \Psi]^{\circ m}(\boldsymbol{\nu}^{\boldsymbol{\alpha}_1}), [\Phi \circ \Psi]^{\circ m}(\boldsymbol{\nu}^{\boldsymbol{\alpha}_0})) \precsim \sup_t \|\nabla_\theta \mathbf{Y}^m\|_E \cdot \mathcal{O}(\gamma^m, C) := \epsilon_m \xrightarrow{m \to \infty} 0. \tag{11}$$

*where a numerical constant* $C$ *is dependent on* $M, b_0, C_1, H_{\boldsymbol{\psi}}, \mathbf{Lip}_{\boldsymbol{b}}, \mathfrak{m}_2, |\mathcal{O}|, e^{-|\mathcal{O}|}, \mathbf{Lip}_W, \mathfrak{h}(\boldsymbol{\alpha}) = \|W_{\boldsymbol{\alpha}}\|_{\mathfrak{g}}$ *is a cut-norm[a] of the proposed graphons (i.e., exponential, cosinusoidal)*

---
[a]Eq. 64 clarifies the explicit upper-bound of the cut-norm for the proposed graphons.

---

Proposition 3.5 reveals two theoretical implications regarding the convergence property. First, the proposed gradient system converges in a distributional sense, as the Wasserstein distance between the populations $([\Phi \circ \Psi]^{\circ m}(\boldsymbol{\nu}^{\boldsymbol{\alpha}_1}) = \boldsymbol{\nu}^{\boldsymbol{\alpha}_{m+1}}$ and $([\Phi \circ \Psi]^{\circ m})(\boldsymbol{\nu}^{\boldsymbol{\alpha}_0}) = \boldsymbol{\nu}^{\boldsymbol{\alpha}_m}$, governed by the gradient norm of the backward dynamics, is expected to decrease as $m$ increases. In other words, $\{\Phi \circ \Psi\}^{\circ m}$ is a Cauchy sequence in $\mathcal{M}$, ensuring the convergent behavior of the proposed training scheme. Second, the proposed gradient system ensures the convergence of the dynamics for the upper bounds $\epsilon_m$. It is important to note that the inequality in Eq (11) is an equivalent expression of the **mean-field Nash** $\epsilon_m$-**equilibrium** described in Definition 3.3. In this context, the neural agent with greater capacity (*i.e.*, a smaller radius $\mathbf{r}_m$ of the metric balls in Eq (49)) further tightens the upper bound. In conclusion, the findings from Proposition 3.5 validate that the proposed gradient system is theoretically sound and efficiently utilizes neural networks to address mean-field games in continuous sequence prediction.

## 4 SAMPLING MEAN-FIELD PREDICTORS

In this section, we propose the numerical algorithm for sampling the proposed mean-field predictors and provide a theoretical analysis of the sample complexity error and the asymptotic convergence of empirical estimation for mean-field predictors.

**Graphon Mean-field Euler Maruyama Scheme.** Inspired by the Euler-Maruyama scheme of Mckean-Vlasov types, we propose an Euler-Maruyama scheme for graphon interacting particle systems to generate a set of mean-field predictors at each time stamp. Alg 1 presents the numerical algorithm for sampling mean-field predictors. Assuming that $\boldsymbol{\alpha}^* := \alpha(\cdot; \theta^*)$ is optimal in the sense of mean-field equilibrium obtained from the operating gradient system of FBSDEs.

---

**Algorithm 1** Sampling Mean-field Continuous Sequence Predictors

---

**while** $t \in \mathbb{T}$ **do**                  ▷ Graphon Mean-field Euler-Maruyama Sampling
    **while** $i \leq N$ **do**
        $\{y_{\mathrm{u}_i}\}_{i \leq N} \sim p(\mathrm{u}, \mathrm{y}), \Delta_t \sim p(\Delta_t), U \sim \mathbf{Unif}(\mathbb{O}), t \sim p(t).$
        $\boldsymbol{\alpha}_i = \alpha(t, \mathbf{X}_i^n; \theta^*), W_{ij} = W_{\boldsymbol{\alpha}_i}(\lceil n\mathrm{u}_i \rceil /n, \lceil n\mathrm{v}_j \rceil /n), \boldsymbol{\psi}_{ij} = \boldsymbol{\psi}_{\boldsymbol{\alpha}_i}\left(\mathbf{X}_i^n(t), \mathbf{X}_j^n(t)\right).$    (12)
        $\mathbf{X}_i^n(t + \Delta_t) = \mathbf{X}_i^n(t) + \frac{1}{n}\sum_j^n W_{ij}\boldsymbol{\psi}_{ij}\Delta_t + \boldsymbol{b}(t, \mathbf{X}_i^n, \boldsymbol{\alpha}_i)\Delta_t + \mathcal{N}(0_d, \sigma_t \Delta_t \mathbf{I}_d).$    (13)
    **end while**                                ▷ Predict Subsequent Future Event
    **if** $t \in \mathbb{T} \setminus \mathbb{O}$ **then**
        $\Lambda_{t+\Delta_t} = \sum_i^K w(U, \lceil n\mathrm{u}_i \rceil /n)\mathbf{X}_i^{n,\boldsymbol{\alpha}_i}(t + \Delta_t) \approx \mathbb{E}_{\mathrm{u} \sim p(\mathrm{u})}\mathbf{X}_{\mathrm{u}}^{\boldsymbol{\alpha}}(t + \Delta_t)$
    **end if**
**end while**

---

Due to the infinite-dimensional nature of the proposed system, sampling mean-field predictors causes inherent complexity errors when applied to finite-dimensional real-world datasets. As the sampled mean-field prediction is expected to approximate its mean-field limit, a natural question arises regarding sample complexity: *How does probability error emerge in relation to sampling complexity?* To rigorously address this, we begin by defining the probabilistic representation of both the sampled and model dynamics as follows:

$$\mathbf{MFPs\ in\ Alg.\ 1} \ : \ \nu_t^N := \frac{1}{N}\sum_i^N \delta_{\mathbf{X}_i^n(t)}, \quad \mathbf{MFPs\ with\ \infty\text{-}order} \ : \ \hat{\mu}_t := \mathbb{E}_{\mathrm{u}\sim p(\mathrm{u})}[\nu_{\mathrm{u}}(t)]. \quad (14)$$

where $\mathbf{X}_i^N(t) \sim \nu_t^N$ is sampled predictors, which can be obtained from implementing the Algorithm 1 and the weighted sum $\Lambda_t$ approximates true collective prediction made by mean-field predictors $\mathbb{E}_{\mathrm{u}}\mathbf{X}_{\mathrm{u}}^{\boldsymbol{\alpha}}(t) \sim \hat{\mu}_t$ in Eq (5). In what it follows, we establish the relation between squared 2-Wasserstein distance and the number of samples $N$, the dimensionality of the data distribution $d$.

---

**Proposition 4.1.** *(Sampling Complexity) For arbitrary* $\mathrm{u} \in \mathcal{O}$*, let* $\nu_t^N, \hat{\mu}_t$ *be probability measures defined in Eq (14). Then, there exist numerical constants* $\mathfrak{c}, \mathfrak{c}_7, \mathfrak{c}_8, \mathfrak{c}_9 > 0, w > 0$ *and* $\kappa > 0$ *such that the probability of squared* 2*-Wasserstein distance can be controlled as follows:*

$$\sup_{t \in \mathbb{T}} \mathbb{P}\left[\mathcal{W}_2^2(\nu_t^N, \hat{\mu}_t) \geq \epsilon\right] \leq \mathfrak{a}\left(\frac{e^{-N\epsilon^2/4\mathfrak{c}}}{\epsilon^2} + \frac{e^{-N\epsilon}}{N}\left(1 - \frac{128\omega\mathfrak{h}(\boldsymbol{\alpha})}{N}\right)^{-d/8} + \frac{1}{72^4 \epsilon\sqrt{N}}\right), \quad (15)$$

$$\mathfrak{a} = \max\left(\mathfrak{c}_9, \frac{2\mathfrak{c}_7^{3/2}}{\kappa}\exp(\mathfrak{c}_4 e^{\frac{1}{2}\mathfrak{c}_1 T})\left(e^{\kappa T} - 1\right), \mathfrak{c}_9 \exp(-4\mathfrak{c}_8)\right). \quad (16)$$

---

The proof primarily draws on the findings presented in Bolley et al. (2007). It is important to note that the result guarantees the proposed system benefits from the *propagation of chaos* (Chaintron & Diez, 2022), validating the asymptotic behavior of the sampled predictions generated by the mean-field predictors.

$$\sup_{t \in \mathbb{T}} \mathcal{W}_2^2\left(\mathbf{Law}\left(\mathbf{X}_{i_1}^n, \cdots, \mathbf{X}_{i_k}^n\right), \ \otimes_{\{j=1,\cdots k\}} \nu_{j/n}(t)\right) \xrightarrow{k \to \infty} 0. \quad (17)$$

Eq. 15 and Eq. 17 and Proposition 8.4 in Appendix align with the intuition that *as the number of predictors $N$ increases (and dimensionality $d$), the sampled dynamics converges more closely to the mean-field limit $\hat{\mu}_t$ and $\nu_{\mathrm{u}}(t)$.* Notably, the right-hand side of the inequality in Eq. (15) is governed by two terms that decay exponentially, and the remaining term decays inversely as a polynomial, both exhibit short-tailed concentration with respect to a number of mean-field predictors.

Moreover, the result demonstrates the advantages of applying mean-field games: Rational individuals (*i.e.*, $\delta_{\mathbf{X}_i^n(t)}$) satisfying Nash equilibrium and conditioned on partial information (*i.e.*, $\mathbf{X}_i^n(0) = \mathrm{y}_{i/n}$) forms a coalition (*i.e.*, $\nu_t^N$), and the group decision is progressively refined to collaboratively solve the continuous sequence prediction problem. **As the coalition size increases, the resulting predictions become progressively more precise and reliable**. In Section 6, we conduct an ablation study to numerically verify these theoretical findings.

Table 1: Mean Squared Errors (MSEs) and Mean Absolute Errors (MAEs) in various continuous sequence prediction tasks. The top and second-top scores in each dataset are highlighted in bold and underlined, respectively. Each metric is scaled by $10^{-2}$.

| Methods | MIT Humanoid Robot | | MIMIC-II | | Beijing Air Quality | |
|---|---|---|---|---|---|---|
| | MSE | MAE | MSE | MAE | MSE | MAE |
| Neural Laplace | 8.11±0.25 | 17.03±0.33 | 7.76±0.04 | 18.70±0.08 | 3.21±0.12 | 11.45±0.23 |
| MaSDEs | 16.51±0.21 | 27.89±0.30 | 8.41±0.06 | 20.67±0.08 | 3.47±0.03 | 13.13±0.07 |
| CRU | 32.08±5.07 | 42.50±3.90 | 13.09±0.31 | 24.68±0.47 | 3.48±0.06 | 12.76±0.19 |
| Latent SDE | 6.01±0.14 | 15.94±0.14 | 8.04±0.02 | 19.63±0.06 | 3.29±0.03 | 11.99±0.07 |
| Neural LSDE | 6.80±0.14 | 16.51±0.08 | 7.93±0.05 | 19.09±0.07 | 3.74±0.04 | 11.98±0.15 |
| CONTIME | 6.88±0.29 | 16.60±0.25 | 12.29±0.14 | 25.26±0.12 | 5.15±0.17 | 15.86±0.27 |
| Contiformer | 5.94±0.23 | 15.29±0.26 | 7.90±0.12 | 19.05±0.18 | 3.25±0.10 | 11.48±0.16 |
| S4 | 5.59±0.16 | 13.98±0.19 | 13.24±0.01 | 24.79±0.30 | 3.95±0.15 | 12.35±0.17 |
| Mamba | 5.21±0.09 | 13.71±0.15 | 13.23±0.02 | 24.76±0.19 | 3.68±0.14 | 11.56±0.24 |
| MFPs (Exp.) | **3.89±0.10** | **11.42±0.14** | 7.51±0.08 | **18.59±0.11** | 3.14±0.07 | 11.45±0.13 |
| MFPs (Cosin.) | 3.91±0.07 | 11.43±0.07 | **7.51±0.06** | 18.60±0.10 | **3.13±0.07** | **11.38±0.08** |

## 5 RELATED WORK

**Neural Differential Equation Models.** In recent years, neural differential equation models have gained attention for their ability to capture the dynamics of complex continuous sequences. Latent ODEs (Rubanova et al., 2019) extend standard RNNs to handle continuous signals by integrating neural ODEs with them. Kidger et al. (2020) introduced differential equation models based on controlled differential equations (Neural CDE) to address a key limitation of neural ODEs, where solutions depend solely on initial conditions and not on subsequent observations. Recently, Contiformer (Chen et al., 2024) was developed, combining neural ODEs and Transformers into a single framework. Another line of research integrates stochasticity by utilizing SDEs, particularly for time-series applications. Latent SDE (Li et al., 2020) encodes sequential data in the latent space using neural SDEs, while MaSDE (Park et al., 2023) employs a concept of stochastic differential games to analyze time series. Koshizuka & Sato (2023) proposed a regularized neural SDE based on the Lagrangian Schrödinger bridge, and Oh et al. (2024) introduced three stable types (classes) of neural SDEs: Langevin-type SDE, Linear Noise SDE, and Geometric SDE.

**Mean-field Principles in Generative Models**. Recent works utilized the mean-field principle to model the infinitely many random particles in high-dimensional data space, where they interact with each other. In Liu et al. (2022), the Schrödinger bridge was incorporated to address mean-field games in order to approximate data distributions for large populations. Park et al. (2024) introduced the concept of propagation of chaos to generate data structures with exchangeable high cardinality such as 3D point clouds.

## 6 EXPERIMENTAL RESULTS

We validate our method on various time-series prediction benchmark datasets, comparing it against several baselines. The details of our experimental settings are as follows:

**Datasets.** In the experiments, we evaluate our results against benchmarks using the following datasets: *(i)* MIT Humanoid Robot (Li et al., 2024), *(ii)* MIMIC-II (Silva et al., 2012), and *(iii)* Beijing Air Quality (Zhang et al., 2017). The MIT Humanoid Robot dataset contains the robot's state trajectories during various activities, such as running, jogging, and stepping in place, with 27 features describing these states. The MIMIC-II dataset, from the PhysioNet Challenge 2012, consists of time series data with 41 features representing the first 48 hours of a patient's ICU admission (e.g., $SaO_2$ and cholesterol levels). The Beijing Air Quality dataset contains time series data for six air pollution indicators, collected from 12 different locations in Beijing. For stable training, we apply either min-max or z-score normalization to each dataset.

**Benchmarks.** Given our focus on continuous sequence modeling, the benchmark baselines consist of various continuous models, including Neural Laplace (Holt et al., 2022), MaSDEs (Park et al., 2023), CRU (Schirmer et al., 2022), Latent SDE (Li et al., 2020), Neural LSDE (Oh et al., 2024), CONTIME (Jhin et al., 2024), and Contiformer (Chen et al., 2024). To further enhance the baselines,

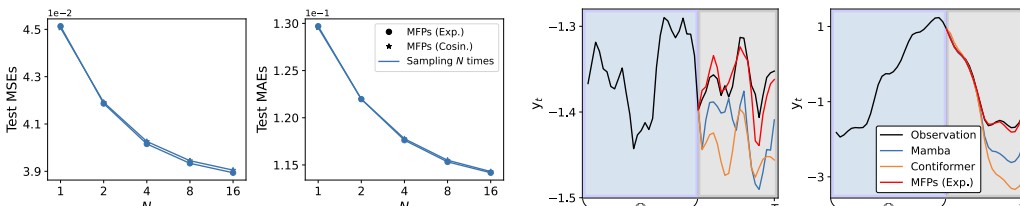

Figure 5: Visualization results on the MIT Humanoid Robot dataset. **(Left)** Sensitivity analysis on the sample complexity. **(Right)** Prediction results compared to representative baselines.

we also incorporate continuous state-space models, such as Mamba (Gu & Dao, 2024) and S4 (Gu et al., 2022). Performance evaluation is carried out using mean squared error (MSE) and mean absolute error (MAE) metrics. Each model is executed five times, with the average scores and standard deviations reported.

**Quantitative Results.** Table 1 presents a performance comparison with benchmark methodologies across three datasets. The results show that the proposed MFPs consistently outperform other benchmarks by significant margins on all datasets. Notably, conventional neural differential equation models perform reasonably well on the MIMIC-II and BAQD datasets, where sequences are irregularly sampled with missing values. However, they exhibit a performance drop on the MIT Humanoid Robot dataset, likely due to their limitations in handling complex spatio-temporal dynamics. In contrast, state-space models excel on the MIT Humanoid Robot dataset but experience a decline in performance on the other two datasets, indicating their limitations in dealing with irregularly sampled sequences. Figure 5 (**right**) illustrates the qualitative prediction results on the MIT Humanoid Robot dataset. As shown, our MFPs deliver superior performance compared to the other models.

**Ablation Study I: Sample Complexity.** To validate the theoretical findings presented in Section 4, we conduct an ablation study to demonstrate the performance improvements as the sampling number $N$ increases. Fig 5 **(Left)** confirms the theoretical findings discussed in Proposition 4.1, indicating that showing that additional performance gains can be realized. It is worth noting that significant performance gains during inference can be achieved by employing multiple mean-field predictors, even after only a single training phase. Since increasing the number of predictors generally results in higher computational costs during the inference, it is essential to select an optimal value for $N$. In all experiments, we consistently set $N = 16$ for balancing efficiency and performance.

**Ablation Study II: Noise Robustness.** We perform a robustness study to assess the impact of non-informative noisy signal (*i.e.*, white noise) interventions in past observations. Specifically, we inject the Gaussian random noises with variance $\sigma_{\text{noise}} = 0.3$ to derive the distributional shift of test continuous-time sequences and corrupt the test data, $\hat{p}(\mathrm{u}, \mathrm{y}) = p(\mathrm{u}, \mathrm{y}) \circledast \mathcal{N}(\mathbf{0}_d, \sigma_{\text{noise}}\mathbf{I}_d)$, where $\circledast$ is a convolution operation. Fig 4 shows a uniform performance degradation (*i.e.*, $\Delta$) with an increasing number of past observations corrupted by non-informative noisy

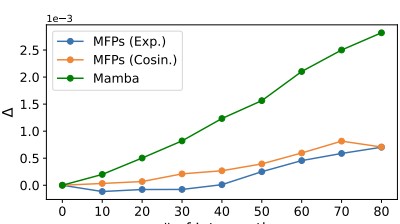

Figure 4: Impact of Noise Intervention

signals. As can be seen, our MFPs exhibit robust performance against noise interventions, as Mamba experiences sharp declines in accuracy under high levels of noise. The coalition, adapted to the original clean sequence $p(\mathrm{u}, \mathrm{y})$, neutralizes the influence of individuals conditioned on noisy signals $\hat{p}(\mathrm{u}, \mathrm{y})$, thereby preserving the Nash equilibrium, resulting the robust generalization performance.

## 7 CONCLUSION

This paper introduces *mean-field continuous sequence predictors*, a novel class of neural SDE model for the efficient generation of continuous sequences, which can possess infinite-order complexity. To capture the complex inductive biases in time-series data, we propose the mean-field dynamics using meticulously designed graphons. We recast the time-series prediction problem as a mean-field game and adopt a fictitious play approach, integrated with a gradient-descent-based method, to leverage the stochastic maximum principle and identify the Nash equilibrium of the system. Both empirical and theoretical results reveal the distinctive features of our MFPs, where the coalition of a continuum of predictors generates accurate predictions and consistently surpasses benchmark performance.

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

# 8 APPENDIX

## 8.1 NOTATIONS AND DEFINITIONS

This section includes brief summary of the mathematical backgrounds, omitted notations and definitions in the manuscript.

**Generalized Wasserstein Distance.** Recall the definition of space for probability measures that consist of generic path measures with finite second moments,

$$\hat{\mathcal{M}} := \{\boldsymbol{\nu} = (\nu_u : u \in \mathcal{O}) \in [C([0, T], \mathbb{R}^d)]^{\mathcal{O}}; \mathrm{u} \mapsto \nu_\mathrm{u} \in \mathcal{P}(C([0, T], \mathbb{R}^d) \text{ is measurable}\},$$

$$\tilde{\mathcal{M}} := \{\boldsymbol{\nu}; \sup_{\mathrm{u} \in \mathcal{O}} \int \|\mathbf{X}_\mathrm{u}(\cdot)\|^2 d\nu_u(\mathbf{X}_\mathrm{u}(\cdot)) < \infty\}.$$

For the arbitrary elements $\boldsymbol{\mu}, \boldsymbol{\nu} \in \mathcal{M} := \hat{\mathcal{M}} \cap \tilde{\mathcal{M}}$, let us consider $\mathcal{M}$ equipped with the generalized 2-Wasserstein metric as

$$\mathcal{W}_{t,\mathcal{M}}(\boldsymbol{\mu}, \boldsymbol{\nu}) := \sup_{u \in \mathcal{O}} \left[ \inf_{\Pi} \mathbb{E} \left( \sup_{s \le t} \|\mathbf{X}_u(s) - \mathbf{Y}_\mathrm{u}(s)\|^2 \right) \right]^{1/2}, \quad \begin{cases} \mathbf{Law}(\mathbf{X}_\mathrm{u}) = P_u^{-1} \circ \boldsymbol{\mu}, \\ \mathbf{Law}(\mathbf{Y}_\mathrm{u}) = P_u^{-1} \circ \boldsymbol{\nu}, \end{cases} \quad (18)$$

where $\Pi$ is a coupling between two probability measures and $P_u$ denotes a canonical projection onto the interval $\mathcal{O}$. Followed by the Kantorovich-Rubinstein duality, definition in Eq (18) can be further modified as

$$L\mathcal{W}_{t,\mathcal{M}}(\boldsymbol{\mu}, \boldsymbol{\nu}) \ge \sup_{\mathrm{u} \in \mathcal{O}} \sup_{f \in \mathbf{Lip}(L)} \left| \int_{\mathbb{R}^d} f d(\mu_{\mathrm{u},t} - \nu_{\mathrm{u},t}) \right|, \quad \boldsymbol{\mu}, \boldsymbol{\nu} \in \mathcal{M}. \quad (19)$$

Note that the inner supremum is taken over a family of $L$-Lipschitz real-valued continuous functions.

**Cut Norm of Graphon**. The cut-norm measures the *discrepancy* between two graphons over all possible cuts of the square of $\mathbb{O}$. Formally, for a graphon $W : \mathbb{O} \times \mathbb{O} \to \mathbb{R}^+$, the cut-norm is defined as:

$$\|W\|_{\mathfrak{g}} := \sup_{A, B \subset \mathcal{O}} \left| \int_{A \times B} W(\mathrm{u}, \mathrm{v}) du dv \right|, \quad (20)$$

where the supremum is taken over all measurable subsets $A$ and $B$. The definition illustrates that the cut-norm quantifies the maximum deviation of $W$ from zero over any rectangle $\mathbb{O}^2$. Given the definition, one defines the metric called *cut distance*:

$$d_{\mathfrak{g}}^q(W_1, W_2) = \|W_1 - W_2\|_{\mathfrak{g}}^q \quad (21)$$

The cut distance measures how close two graphons are after optimally aligning their domains. If the cut distance between two graphons $W_1$ and $W_2$ is small, the graphs they represent are structurally similar.

**(Exponential AM-GM Inequality).** For the arbitrary random variables $X, Y$ and positive constants $a, b > 0$, the expectation can be decomposed as follows:

$$\mathbb{E}[\exp(aX^2 + bY^2)] \le \left( 2\mathbb{E}[\exp(2X^2)] \right)^{1/2} \left( 2\mathbb{E}[\exp(2X^2)] \right)^{1/2}. \quad (22)$$

**(Arithmetic AM-GM Inequality).** For arbitrary positive constants $x, y, w > 0$, we have

$$xy \le \omega x + \frac{1}{4\omega} y. \quad (23)$$

## 8.2 ASSUMPTION

Without additional information, we make the following assumptions in this paper.

1. (**H1**). There exists a finite collection of intervals $\{O_k; k \in \{1, \cdots, N\}\}$ for arbitrary $N \in \mathbb{N}^+$ such that $\cup_k^N O_k = \mathbb{O}$. Then we assume the following:

   - For each $k$, the initial datum of the graphon system is set with the data distribution $\nu_{\mathrm{u}}$: $O_k \ni \mathrm{u} \mapsto \mu_{\mathrm{u}}(0) \coloneqq \nu_{\mathrm{u}} \in \mathcal{P}_2$, where the mapping assigns to independent measures.

2. (**H2**). For each $k$ and $O_k \ni \mathrm{u}$, there exists a constant $C_1$ such that we have probability $\nu_{u,s}[\sup_{x \in \mathbb{R}^d \setminus \mathbf{Y}(\omega)} \|\mathrm{x} - \mathbf{Y}\|^{-p} \leq C_1]$ almost surely for all $p \in \mathbb{N}^+$, and the second moment (*i.e.*, $\mathfrak{m}_2$) of $\nu_{u,s}$ is bounded.

3. (**H3**). The Lipschitz constants of the functions in modeling of graphons $W_{(\cdot)} : \mathbb{L}_2(\nu_{\mathrm{u}}(t)) \supset \mathbb{A} \to \mathbb{R}^+$ are bounded above. The parameterized Markovian feedback controls are Lipschitz in parameters:

$$|W_{(\cdot)}(\boldsymbol{\alpha}) - W_{(\cdot)}(\boldsymbol{\beta})| \leq \mathbf{Lip}_W \|\boldsymbol{\alpha} - \boldsymbol{\beta}\|_{\nu_{\mathrm{u}}(t)}, \tag{24}$$

$$\{\|\boldsymbol{\alpha} - \boldsymbol{\beta}\|_{\nu_{\mathrm{u}}(t)}, \|\alpha(t, \mathrm{x}, \theta_{\boldsymbol{\alpha}}) - \alpha(t, \mathrm{x}, \theta_{\boldsymbol{\beta}})\|\} \leq \mathbf{Lip}_{\theta} \|\theta_{\boldsymbol{\alpha}} - \theta_{\boldsymbol{\beta}}\|_E \tag{25}$$

   The drift function is Lipschitz continuous and dissipative, ensuring that the constant $\mathfrak{c}_1$ is well-defined.

$$\|(\boldsymbol{b}, \boldsymbol{b}_W)(t, \mathrm{x}, \boldsymbol{\alpha}) - (\boldsymbol{b}, \boldsymbol{b}_W)(t, \mathrm{y}, \boldsymbol{\beta})\| \leq \mathbf{Lip}_{\boldsymbol{b}}(\|\mathrm{x} - \mathrm{y}\|_E + \|\boldsymbol{\alpha} - \boldsymbol{\beta}\|_{\nu_{\mathrm{u}}(t)}). \tag{26}$$

$$\mathfrak{c}_1 \coloneqq \inf_{\mathrm{x},\mathrm{y}} -(\mathrm{x} - \mathrm{y}) \cdot [(\boldsymbol{b}, \boldsymbol{b}_W)(\mathrm{x}) - (\boldsymbol{b}, \boldsymbol{b}_W)(\mathrm{y})] / \|\mathrm{x} - \mathrm{y}\|_E^2 \tag{27}$$

4. (**H4**). The maximal rank of embedding of neural agents in $\mathbb{A}$ is $d'$.

$$\mathbb{T} \times \mathbb{R}^d \times \Theta \mapsto \boldsymbol{\alpha} \in \mathbb{A} \hookrightarrow \mathbb{L}_2(\boldsymbol{\nu}). \tag{28}$$

## 8.3 PROOFS

## 8.4 STOCHASTIC OPTIMAL CONTROL, MEAN-FIELD FBSDEs

Before presenting the main proofs, this section offers a detailed analysis of how the proposed mean-field games can be formulated.

**Weak Formulation of Mean-field Games.** We start by explicating on the rigorous definition of forward mean-field dynamic in Eq. (1) cost functional in Eq. (5) and gradient system of FBSDEs in Propsoition 3.4, followed by a brief summary of how forward-backward SDEs are formulated in the context of stochastic optimal control problems. To this end, let us first define the primitive process $\bar{\mathbf{X}}_t$, which solves the following SDE for a fixed label u:

$$d\bar{\mathbf{X}}_{\mathrm{u}}(t) = \sigma_t dB_t^{\mathrm{u}}, \quad \bar{\mathbf{X}}_0(t) = \mathrm{y}_t. \tag{29}$$

where $B_t^{\mathrm{u}}$ is a Brownian motion under probability measure $\mathbb{P}$. Given the square of volatility term $\sigma_t^2$ is bounded below some constant, we introduce the probability measure $\mathbb{P}^{\boldsymbol{\mu}, \boldsymbol{\alpha}}$, which can be derived by the following Radon-Nikodym derivative:

$$\frac{d\mathbb{P}^{\boldsymbol{\mu}, \boldsymbol{\alpha}}}{d\mathbb{P}} = \mathcal{E}\left(\int_0^{(\cdot)} \sigma_t^{-1} \left(\boldsymbol{b}_W(\bar{\mathbf{X}}_{\mathrm{u}}(t), \boldsymbol{\nu}, \boldsymbol{\alpha}) + \boldsymbol{b}(t, \bar{\mathbf{X}}_{\mathrm{u}}(t), \boldsymbol{\alpha})\right) \cdot dB_t^{\mathrm{u}}\right)\bigg|_{t=T}. \tag{30}$$

where $\mathcal{E}$ denotes a Doléans-Dade exponential of a martingale. Applying Girsanov's theorem, we have the Brownian motion $W^{\boldsymbol{\mu}, \boldsymbol{\alpha}}$ under the probability measure $\mathbb{P}^{\boldsymbol{\mu}, \boldsymbol{\alpha}}$:

$$W_t^{\mathrm{u}, \boldsymbol{\mu}, \boldsymbol{\alpha}} = B_t^{\mathrm{u}} - \int_{\mathbb{T}} \sigma_s^{-1} \left(\boldsymbol{b}_W(\bar{\mathbf{X}}_{\mathrm{u}}(s), \boldsymbol{\nu}, \boldsymbol{\alpha}) + \boldsymbol{b}(s, \bar{\mathbf{X}}_{\mathrm{u}}(s), \boldsymbol{\alpha})\right) ds. \tag{31}$$

Then, the primitive process can be rewritten as follows almost surely $\mathbb{P}^{\boldsymbol{\mu}, \boldsymbol{\alpha}}$,

$$d\bar{\mathbf{X}}_{\mathrm{u}}(t) = \left(\boldsymbol{b}_W(\bar{\mathbf{X}}_{\mathrm{u}}(t), \boldsymbol{\nu}, \boldsymbol{\alpha}) + \boldsymbol{b}(t, \bar{\mathbf{X}}_{\mathrm{u}}(t), \boldsymbol{\alpha})\right) dt + \sigma_t dW_t^{\mathrm{u}, \boldsymbol{\mu}, \boldsymbol{\alpha}}. \tag{32}$$

By suppressing the objects in upper-scripts for simplicity, with the notation $W_t^{\mathrm{u}} = W_t^{\mathrm{u},\boldsymbol{\mu},\boldsymbol{\alpha}}$ and $\mathbf{X}_{\mathrm{u}}(t) = \bar{\mathbf{X}}_{\mathrm{u}}(t)$, one can recover the original mean-field forward SDE defined in Eq (1). Note that this formulation reveals that the process $\bar{\mathbf{X}}_{\mathrm{u}}(t)$ is a weak solution under $\mathbb{P}^{\boldsymbol{\mu},\boldsymbol{\alpha}}$, and the cost functional can be posed as follows:

$$\mathcal{J}(\boldsymbol{\nu}^{\boldsymbol{\alpha}}, \boldsymbol{\alpha}) = \int_{\mathbb{T}} \mathbb{E}_{\boldsymbol{\alpha},\boldsymbol{\nu}} \left[ \|\mathbb{E}_{\mathrm{u}\sim p(\mathrm{u})} \mathbf{X}_{\mathrm{u}}^{\boldsymbol{\alpha}}(t) - y_t\|_E^2 + \mathbf{G}(\mathbf{X}_{\mathrm{u}}^{\boldsymbol{\alpha}}(T), \boldsymbol{\nu}^{\boldsymbol{\alpha}}) \right] dt, \tag{33}$$

where the expectation $\mathbb{E}_{\boldsymbol{\alpha},\boldsymbol{\nu}}$ is taken with respect to $\mathbb{P}^{\boldsymbol{\mu},\boldsymbol{\alpha}}$. Note that the cost functional in Eq. (5) is an alternative form of Eq. (33). Next, we reformulate the approximation of mean-field games with graphon in the probabilistic sense. Let $\boldsymbol{\alpha} = \alpha(t,\mathrm{x};\theta) := \hat{\alpha}(t,\mathrm{x},\bar{\mu},\bar{\mathfrak{e}}) := \hat{\boldsymbol{\alpha}}$ be an extended control with fixed arguments $\bar{\mu}, \bar{\mathfrak{e}}$. For the fixed $\boldsymbol{\nu}_u^{\boldsymbol{\alpha}^*}$ a.e., $\mathrm{u} \sim \mathrm{v}_{\mathrm{Unif}}$ associated with the optimal control $\hat{\boldsymbol{\alpha}}^*$, let us consider a Hamiltonian-Jacobi-Bellman equation (HJBE), having a classical value function $\mathcal{V}$:

$$\partial_t \mathcal{V}(t,\mathrm{x}) + \frac{1}{2}\mathbf{Tr}[\sigma_t^2 \partial_{\mathrm{xx}}^2 \mathcal{V}(t,\mathrm{x})] + H\left(t,\mathrm{x}, \boldsymbol{\nu}_u^{\boldsymbol{\alpha}^*}, \partial_{\mathrm{x}}\mathcal{V}(t,\mathrm{x}), \hat{\boldsymbol{\alpha}}^*(t,\mathrm{x},\boldsymbol{\nu}_u^{\boldsymbol{\alpha}^*}, \partial_{\mathrm{x}}\mathcal{V}(t,\mathrm{x}))\right) = 0, \tag{34}$$

Then, forward-backward SDEs associated with the Hamiltonian system in (34) can be described in the Proposition 8.1:

**Proposition 8.1.** *(Weak Formulation: Forward-Backward SDEs I) (Carmona & Delarue, 2013) For the fixed flow of measures $\nu_u(\cdot) : \mathbb{T} \to \mathcal{P}_2$ and the fixed label $\mathrm{u}$, let $(\mathbf{X}_{\mathrm{u}}(t), \mathbf{Y}_{\mathrm{u}}(t), \mathbf{Z}_{\mathrm{u}}(t))$ be a family of processes that solves forward-backward stochastic differential equations with respect to the proposed graphon system in Eq (1) given as follows:*

$$d\mathbf{X}_{\mathrm{u}}(t) = (\boldsymbol{b}_W(\mathbf{X}_u(t), \nu_u, \hat{\boldsymbol{\alpha}}^*) + \boldsymbol{b}(t, \mathbf{X}_u(t), \hat{\boldsymbol{\alpha}}^*))\,dt + \sigma_t dW_t^{\mathrm{u}}, \tag{35}$$

$$d\mathbf{Y}_{\mathrm{u}}(t) = -H(t, \mathbf{X}_u(t), \mathbf{Y}_u(t), \nu_u, \hat{\boldsymbol{\alpha}}^*)dt + \mathbf{Z}_{\mathrm{u}}(t) \cdot dW_t^{\mathrm{u}}. \tag{36}$$

*where $\boldsymbol{b}_W(\mathrm{x}, \nu, \alpha) := \langle \mathbb{W}_{\boldsymbol{\alpha}}[\nu](\mathrm{u}), \boldsymbol{\psi} \rangle(\mathrm{x}, \alpha)$ is the graphon interaction term, and terminal constraint is given as $\mathbf{Y}_{\mathrm{u}}(T) = \mathbf{G}(\mathbf{X}_T, \boldsymbol{\nu}_T)$. Then, under the mild assumption (e.g., smooth boundness of $\partial_x \mathcal{V}$ and $\partial_{xx}\mathcal{V}$), there exist solutions of stochastic optimal control of the following minimization problem:*

$$\inf_{\boldsymbol{\alpha} \in \mathbb{A}} \mathcal{J}(\boldsymbol{\nu}^{\boldsymbol{\alpha}}, \boldsymbol{\alpha}) = \mathbf{Y}_{\mathrm{u}}(0). \tag{37}$$

For the closed Markovian control such as neural control introduced in Section 2, the solution to adjoint process $\mathbf{Z}_{\mathrm{u}}(t)$ can be defined as stated in Definition 3.4. By rewriting forward-backward SDEs in Eq (35) and Eq (36) for non-optimal neural controls $\boldsymbol{\alpha}$ (*i.e.*, neural networks) which are updated via gradient descent, we can recover the proposed gradeint system of FBSDEs in Definition 3.2.

### 8.4.1 ANALYSIS ON STOCHASTIC OPTIMALITY AND CONVERGENCE

**Stochastic Optimality.** In the following, we introduce the second type of forward-backward SDEs, which is based on the principles of stochastic maximum principle:

**Proposition 8.2.** *(Stochastic Maximum Principle: Forward-Backward SDEs II) (Bensoussan et al., 2013) For the fixed flow of measures $\nu_u(\cdot) : \mathbb{T} \to \mathcal{P}_2$ and the fixed label $\mathrm{u}$, let $(\mathbf{X}_{\mathrm{u}}(t), \mathbf{Y}_{\mathrm{u}}^{MP}(t), \mathbf{Z}_{\mathrm{u}}^{MP}(t))$ be a family of processes that solves forward-backward stochastic differential equations with respect to the proposed graphon system in Eq (1) given as follows:*

$$d\mathbf{X}_{\mathrm{u}}(t) = (\boldsymbol{b}_W(\mathbf{X}_u(t), \nu_u, \hat{\boldsymbol{\alpha}}^*) + \boldsymbol{b}(t, \mathbf{X}_u(t), \hat{\boldsymbol{\alpha}}^*))\,dt + \sigma_t dW_t^{\mathrm{u}},$$

$$d\mathbf{Y}_{\mathrm{u}}^{MP}(t) = -\partial_{\mathrm{x}} H(t, \mathbf{X}_{\mathrm{u}}(t), \mathbf{Y}_{\mathrm{u}}^{MP}(t), \nu_u, \hat{\boldsymbol{\alpha}}^*)dt + \mathbf{Z}_t^{MP} \cdot dW_t^{\mathrm{u}}.$$

*For the progressively measurable admissible Markovian neural control $\boldsymbol{\beta}$ under the mild assumption (e.g., smooth boundness of $\partial_x \mathcal{V}$ and $\partial_{xx}\mathcal{V}$), there exists a constant $\tau_{SMP} > 0$ such that the following inequality holds:*

$$\mathcal{J}(\boldsymbol{\nu}^{\hat{\boldsymbol{\alpha}}^*}, \hat{\boldsymbol{\alpha}}^*) + \tau_{SMP} \int_{\mathbb{T}} \|\hat{\boldsymbol{\alpha}}^* - \boldsymbol{\beta}\|_{\boldsymbol{\nu}} dt \leq \mathcal{J}(\boldsymbol{\nu}^{\boldsymbol{\beta}}, \boldsymbol{\beta}). \tag{38}$$

*Remark.* Note that the backward dynamics $\mathbf{Y}_{\mathrm{u}}^{\mathbf{MP}}$ differs from the original backward dynamics $\mathbf{Y}_{\mathrm{u}}$ in Definition (3.4) as the dynamics is designed to be associated with *Pontryagin stochastic maximum principle*. This principle plays a central role in the proof of Proposition 8.3, demonstrating the stochastic optimality of neural agents in the following section.

In what it follows, we demonstrate that the stochastic optimality of the proposed gradient system can be guaranteed under the specific conditions required for constructing the control set in Prop 8.3.

**Proposition 8.3.** *(Maximum Principle of Graphon Mean-field System) Assume that there exists a constant $K_H$ such that $\|\partial_{\boldsymbol{\alpha}}\|\boldsymbol{H}\|_E\|_{\infty,\boldsymbol{\nu}} \leq K_H$. Then, there exists a convex set of admissible neural agents $\boldsymbol{\alpha}_m \in \mathbb{A}$ such that the following relation holds:*

$$D_{\boldsymbol{\alpha}}\mathcal{J}(\boldsymbol{\nu}^{\boldsymbol{\alpha}_m}, \boldsymbol{\alpha}_m) := \lim_{\varepsilon \to 0} \frac{d}{d\varepsilon}\mathcal{J}\left[\boldsymbol{\alpha}_m + \varepsilon(\boldsymbol{\alpha}_m - \boldsymbol{\alpha}_{m-1})\right] \xrightarrow{m \to \infty} 0. \tag{39}$$

*Furthermore, the sequence of control profile $\{\boldsymbol{\alpha}_m\}$ leads to the minimization of the stochastic Hamiltonian system in terms of **Pontryagin maximum principle**:*

$$\lim_{m \to \infty} H(t, \mathbf{X}_{\mathrm{u}}^m(t), \mathbf{Y}_{\mathrm{u}}^{m,\mathbf{MP}}(t), \nu_{\mathrm{u}}, \boldsymbol{\alpha}_m) = \inf_{\boldsymbol{\alpha} \in \mathbb{A}} H(t, \mathbf{X}_{\mathrm{u}}(t), \mathbf{Y}_{\mathrm{u}}^{MP}(t), \nu_{\mathrm{u}}, \boldsymbol{\alpha}), \ dt \otimes d\mathbb{P} - a.e. \tag{40}$$

*where the population is set to $\nu_{\mathrm{u}} = \Psi(\boldsymbol{\nu}^{\boldsymbol{\alpha}_{m-1}}) := \Psi(\nu_{\mathrm{u}}^{\boldsymbol{\alpha}_{m-1}})$. In other words, the value function can be derived by the proposed gradient system of FBSDEs:*

$$\mathcal{V} := \inf_{\boldsymbol{\alpha} \in \mathbb{A}} \mathcal{J}(\boldsymbol{\nu}^{\boldsymbol{\alpha}}, \boldsymbol{\alpha}) = \lim_{m \to \infty} \mathcal{J}(\boldsymbol{\nu}^{\boldsymbol{\alpha}_m}, \boldsymbol{\alpha}_m). \tag{41}$$

*Proof.* We divide the proof into two separate steps.

**1. Computation of Gâteaux derivative $D_{\boldsymbol{\alpha}}\mathcal{J}$.** The aim of the first step is to provide an explicit computation of the Gâteaux derivative of cost functional (value function) with respect to the neural agent. To achieve this, we introduce the variation equation $\mathrm{i}_{\mathrm{u}}$ and its associated gradient system of SDEs with fixed $\boldsymbol{\beta}$:

$$d\mathbf{Y}_{\mathrm{u}}^{m,\mathbf{MP}}(t) = -\partial_{\mathrm{x}}H(t, \mathbf{X}_{\mathrm{u}}^m(t), \mathbf{Y}_{\mathrm{u}}^{m,\mathbf{MP}}(t), \nu_{\mathrm{u}}, \hat{\boldsymbol{\alpha}}_m)dt + \mathbf{Z}_t^{m,\mathbf{MP}} \cdot dW_t^{\mathrm{u}}, \tag{42}$$

$$d\mathrm{i}_{\mathrm{u}}(t) = [(\partial_{\mathrm{x}}\boldsymbol{b}_W + \partial_{\mathrm{x}}\boldsymbol{b})\mathrm{i}_{\mathrm{u}}(t)]dt + [(\partial_{\boldsymbol{\alpha}}\boldsymbol{b}_W + \partial_{\boldsymbol{\alpha}}\boldsymbol{b})\boldsymbol{\beta}_m]dt, \tag{43}$$

$$d\mathrm{j}_{\mathrm{u}}(t) := d[\mathrm{i}_{\mathrm{u}}(t) \cdot \mathbf{Y}_{\mathrm{u}}^{m,\mathbf{MP}}(t)]dt \in \mathbb{R}^d. \tag{44}$$

Let $\Upsilon_{\boldsymbol{\alpha}}(m, \epsilon) := \boldsymbol{\alpha}_m + \epsilon\boldsymbol{\beta}_m$ represent the infinitesimal changes of the admissible neural agent $\boldsymbol{\alpha}^m$ in the direction of $\boldsymbol{\beta}_m := \boldsymbol{\alpha}_{m-1} - \boldsymbol{\alpha}_m$. To feasibly select the convex combination $\Upsilon_{\boldsymbol{\alpha}}(m, \epsilon)$ for any $m$ and $\epsilon \in [0, 1]$, both neural agents need to lie within some convex set $\mathbb{A}_m$. For now, we assume that there exists a convex set $\mathbb{A}_m$ that includes $\boldsymbol{\alpha}$ and $\boldsymbol{\beta}$. The explicit form of this set will be clarified in the subsequent step. Given the definition, we compute the derivative as follows:

$$D_{\boldsymbol{\alpha}}\mathcal{J}(\boldsymbol{\nu}^{\boldsymbol{\alpha}_m}, \boldsymbol{\alpha}_m) = \frac{d}{d\epsilon}\mathcal{J}(\boldsymbol{\nu}^{\Upsilon_{\boldsymbol{\alpha}}(m,\epsilon)}, \Upsilon_{\boldsymbol{\alpha}}(m, \epsilon))|_{\epsilon=0}$$
$$= \mathbb{E}\left[\int_{\mathbb{T}}[\mathrm{i}_{\mathrm{u}}(t)\partial_{\mathrm{x}}f + \boldsymbol{\beta}_m\partial_{\boldsymbol{\alpha}}f]dt + \mathrm{i}_{\mathrm{u}}(T)\partial_x\boldsymbol{G}\right], \tag{45}$$

where we denote $f(t, \mathrm{x}, \alpha) = \|\mathbb{E}_{\mathrm{u}}[\mathrm{x}^{\alpha}(t)] - y_t\|^2$. While $\mathrm{i}_u(T)\partial_x\mathbf{G}$ can be identified with $\mathrm{j}_u(T)$, we apply the product rule to the third dynamics $d\mathrm{j}_{\mathrm{u}}$ in Eq (44) to have variational form to induce $\mathrm{j}_u(T)$:

$$d\mathrm{j}_{\mathrm{u}}(t) = [\mathbf{Y}_{\mathrm{u}}(t) \cdot d\mathrm{i}_{\mathrm{u}}(t)]dt + [\mathrm{i}_{\mathrm{u}}(t) \cdot d\mathbf{Y}_{\mathrm{u}}^{\mathbf{MP}}(t)]dt + \mathbf{Tr}[d\mathbf{Y}_{\mathrm{u}}^{\mathbf{MP}}(t) \otimes d\mathrm{i}_{\mathrm{u}}(t)]$$
$$= \int_0^T \mathbf{Y}_{\mathrm{u}}^{\mathbf{MP}}(t) \cdot (\partial_{\mathrm{x}}\boldsymbol{b}_W + \partial_{\mathrm{x}}\boldsymbol{b})\boldsymbol{\beta}_m + \mathbf{Y}_{\mathrm{u}}^{\mathbf{MP}}(t) \cdot (\partial_{\boldsymbol{\alpha}}\boldsymbol{b}_W + \partial_{\boldsymbol{\alpha}}\boldsymbol{b})\mathrm{i}_{\mathrm{u}}(t)dt \tag{46}$$
$$= \int_0^T \partial_{\mathrm{x}}\mathbf{G} \cdot (\partial_{\mathrm{x}}\boldsymbol{b}_W + \partial_{\mathrm{x}}\boldsymbol{b})\boldsymbol{\beta}_m + \partial_{\mathrm{x}}\mathbf{G} \cdot (\partial_{\boldsymbol{\alpha}}\boldsymbol{b}_W + \partial_{\boldsymbol{\alpha}}\boldsymbol{b})\mathrm{i}_{\mathrm{u}}(t)dt.$$

Combining Eq (45) with Eq (46), and Cauchy–Schwarz inequality gives explicit form for the Gâteaux derivative of objective functional.

$$
\begin{aligned}
D_{\boldsymbol{\alpha}}\mathcal{J}(\Upsilon_{\boldsymbol{\alpha}}(m,\epsilon)) &= \mathbb{E}\left[\int_{\mathbb{T}}\partial_{\boldsymbol{\alpha}}H(t,\mathbf{X}_{\mathrm{u}}^m,\mathbf{Y}_{\mathrm{u}}^{m,\mathbf{MP}}(t),\Psi(\boldsymbol{\nu}^{\boldsymbol{\alpha}^{m-1}}),\boldsymbol{\alpha}_m)dt\cdot\boldsymbol{\beta}_m\right] \\
&\leq \mathbb{E}\left[\int_{\mathbb{T}}\|\partial_{\boldsymbol{\alpha}}H(t,\mathbf{X}_{\mathrm{u}}^m,\mathbf{Y}_{\mathrm{u}}^{m,\mathbf{MP}}(t),\Psi(\boldsymbol{\nu}^{\boldsymbol{\alpha}^{m-1}}),\boldsymbol{\alpha}_m)\|_E\cdot\|\boldsymbol{\beta}_m\|_E dt\right] \\
&\leq \left|\left\|\partial_{\boldsymbol{\alpha}}\mathbf{H}^m\|_E\right\|\right|_\infty\cdot\left|\left\|\boldsymbol{\beta}_m\|_E\right\|\right|_1,
\end{aligned}
\tag{47}
$$

where $\|\cdot\|_p$ denotes $L_p$-norm, and the last inequality is obtained by applying Hölder's inequality with the conjugate pair $(p=\infty, q=1)$. Then, we have

$$
\begin{aligned}
\left|\left\|\boldsymbol{\beta}_m\|_E\right\|\right|_1 &= \left|\left\|\boldsymbol{\alpha}_m-\boldsymbol{\alpha}_{m-1}\|_E\right\|\right|_1 := \left|\left\|\alpha(t,\mathbf{X}_{\mathrm{u}}^m,\theta^m)-\alpha(t,\mathbf{X}_{\mathrm{u}}^m,\theta^{m-1})\|_E\right\|\right|_1 \\
&\leq \gamma^{m-1}\mathbf{Lip}_\alpha\mathbb{E}\delta_\theta\mathbf{Y}^{m-1}, \qquad \delta_\theta\mathbf{Y}^{m-1} := \|\nabla_\theta\mathbf{Y}_{\mathrm{u}}^{m-1,\boldsymbol{\alpha}^{m-1}}(t)\|_E.
\end{aligned}
\tag{48}
$$

**2. Construction of $\mathbb{A}$.** Next, we define the explicit form of the control set $\mathbb{A}_m$. The constructed control set must meet two conditions: (1) it must be convex, and (2) the right-hand side of the inequality in Eq (47) must converge. For properly dealing with the first condition, let us consider a metric ball $\mathbf{B}_m$ in $L_1$ space as follows:

$$
\mathbf{B}_m := B(\boldsymbol{\alpha}_{m-1},\boldsymbol{r}_m)\in\mathbb{L}_1,
\tag{49}
$$

$$
\boldsymbol{r}_m := r_{\mathrm{u},t,m} = \varepsilon\gamma^{m-1}\mathbf{Lip}_\alpha\delta_\theta Y_{\mathrm{u}}^{m-1}(t), \quad \varepsilon\in[0,1].
\tag{50}
$$

Since any arbitrary metric ball is convex and the calculated reverse direction of gradient guarantees local minimum at each stage, the setup of the proposed metric ball ensures the well-definedness of Gâteaux derivative in Eq (47) and local optimality at each stage $m$.

Let $\lambda_{\max}^m(\boldsymbol{\alpha})$ be an eigenvalue with respect to the principal direction of Hessian for cost functional, $i.e.$, $\mathbf{Hess}_\theta\mathcal{J}(\boldsymbol{\nu}^{\boldsymbol{\alpha}},\boldsymbol{\alpha}(\cdot;\theta))$. Consider another control set $\mathbb{C}_m := \{\boldsymbol{\alpha}_{m-1};\lambda_{\max}^{m-1}(\boldsymbol{\alpha}_{m-1})\leq(\gamma^{m-1})^{-1}\}$. The conventional analysis of gradient descent gives the following inequality on $\mathbb{C}_m$:

$$
\mathbb{E}\mathbf{Y}^{m,\boldsymbol{\alpha}_m}\leq\mathbb{E}\mathbf{Y}^{m-1,\boldsymbol{\alpha}_{m-1}}-\frac{1}{2}\left(2\gamma^{m-1}-(\gamma^{m-1})^2\lambda_{\max}^{m-1}(\boldsymbol{\alpha}_{m-1})\right)\left(\mathbb{E}\delta_\theta\mathbf{Y}^{m-1}\right)^2.
\tag{51}
$$

While the second term in right-hand side of Eq (51) is non-negative, the sequence of expectations for the backward dynamics is non-increasing, demonstrating that $\lim_{m\to\infty}D_{\boldsymbol{\alpha}}\mathcal{J}\leq\lim_{m\to\infty}\mathbb{E}\delta_\theta\mathbf{Y}^{m-1}=0$ when the infinite sequence $\{\boldsymbol{\alpha}_m\}$ lies within $\lim_{m\to\infty}\mathbb{C}_m$. To inherit aforementioned properties lying in both control profiles for all $m$, we define $\mathbb{A}_m := \bigsqcup_{m\geq\mathrm{m}}(\mathbb{B}_\mathrm{m}\cap\mathbb{C}_\mathrm{m})$, where $\mathbb{A}=\lim_{m\to\infty}\mathbb{A}_m$. The result directly follows from findings in the stochastic maximum principle (SMP) (Carmona et al., 2018; Bensoussan et al., 2013), ensuring the equivalence of the following relation:

$$
\mathbb{E}\partial_{\boldsymbol{\alpha}}\mathbf{H}(\cdot,\boldsymbol{\alpha}^*)\cdot\boldsymbol{\beta}_m=0 \quad\longleftrightarrow\quad \boldsymbol{\alpha}^*=\arg\inf_{\boldsymbol{\alpha}\in\mathbb{A}}\mathbf{H}(\cdot,\boldsymbol{\alpha}).
\tag{52}
$$

Note that this equivalence relation is applicable only when $\mathbb{A}$ is constructed in the manner previously specified. $\qquad\square$

### 8.5 CONVERGENCE OF GRADIENT SYSTEM OF FBSDEs, MEAN-FIELD EQUILIBRIUM

As we have formally defined the stochastic optimal control problem and established the corresponding optimality conditions, this section delves into the detailed rationale of how the proposed gradient descent-based FBSDEs achieve the Nash equilibrium. We will prove Proposition 3.5 through the following steps:

1. For the arbitrary probability measures (*i.e.*, $\boldsymbol{\mu}^{\boldsymbol{\beta}}, \boldsymbol{\nu}^{\boldsymbol{\alpha}}$) associated with fixed Markovian controls $\boldsymbol{\alpha}$ and $\boldsymbol{\beta}$, we first establish that the upper bounds of the generalized Wasserstein distance remain stable when two measure-valued operators $\Phi$ and $\Psi$ are composed repeatedly:

$$\mathcal{W}_{t,\mathcal{M}}([\Phi \circ \Psi]^{\circ m}(\boldsymbol{\mu}^{\boldsymbol{\beta}}), [\Phi \circ \Psi]^{\circ m}(\boldsymbol{\nu}^{\boldsymbol{\alpha}})) \xrightarrow{m \geq M} 0. \tag{53}$$

2. Consequently, we reparameterize reference measures ($\boldsymbol{\mu}^{\boldsymbol{\beta}}, \boldsymbol{\nu}^{\boldsymbol{\alpha}}$) with the laws of inferred mean-field forward dynamics in Eq 1 at subsequent stages (*i.e.*, $\boldsymbol{\nu}^{\boldsymbol{\alpha}^m}, \boldsymbol{\nu}^{\boldsymbol{\alpha}^{m+1}}$), proving the convergence towards mean-field Nash equilibrium.

**Proposition 3.5.** *With the assumptions explored in the previous proof, for the fixed label* $\mathrm{u} \sim p(\mathrm{u})$, *the* $m$-*fold of composition* $\Phi \circ \Psi$ *induces convergent behavior of generalized Wasserstein distance:*

$$\mathcal{W}_2([\Phi \circ \Psi]^{\circ m}(\boldsymbol{\nu}^{\boldsymbol{\alpha}_1}), [\Phi \circ \Psi]^{\circ m}(\boldsymbol{\nu}^{\boldsymbol{\alpha}_0}))^2 \leq \sup_{t \in \mathbb{T}} \mathcal{W}_{t,\mathcal{M}}([\Phi \circ \Psi]^{\circ m}(\boldsymbol{\nu}^{\boldsymbol{\alpha}_1}), [\Phi \circ \Psi]^{\circ m}(\boldsymbol{\nu}^{\boldsymbol{\alpha}_0}))^2$$

$$\leq \lim_{M \to \infty} \frac{C(T)^M (\sup_t \sup_m \mathbf{r}_m)^M - 1}{C(T)(\sup_t \sup_m \mathbf{r}_m) - 1} + \frac{(C'T)^M}{M!} \sup_{t \in \mathbb{T}} \mathcal{W}_{t,\mathcal{M}}(\boldsymbol{\nu}^{\boldsymbol{\alpha}_1}, \boldsymbol{\nu}^{\boldsymbol{\alpha}_0})^2 \xrightarrow{M \to \infty} 0. \tag{54}$$

*where a numerical constant* $C$ *is dependent on* $b_0, C_1, H_{\boldsymbol{\psi}}, \mathbf{Lip}_{\boldsymbol{b}}, \mathfrak{m}_2, |\mathcal{O}|, e^{-|\mathcal{O}|}, \mathfrak{h}, \mathbf{Lip}_W$. *In other words,* $[\Phi \circ \Psi]^{\circ m}$ *is a Cauchy sequence on* $\mathcal{M}$, *and the proposed gradient system converges.*

*Proof.* Recall the definition of controlled graphon system that the particle dynamics at time $t$ with distinctive controls $\boldsymbol{\alpha}$ and $\boldsymbol{\beta}$ can be presented as follows:

$$\mathbf{X}_{\mathrm{u}}^{\boldsymbol{\nu},\boldsymbol{\alpha}}(t) = \mathbf{X}_{\mathrm{u}}^{\boldsymbol{\nu},\boldsymbol{\alpha}}(0) + \int_0^t \langle \mathbb{W}_{\boldsymbol{\alpha}}[\nu_{\mathrm{v},s}], \boldsymbol{\psi} \rangle (\mathbf{X}_u^{\boldsymbol{\alpha}}(s)) ds + \int_0^t \boldsymbol{b}(s, \mathbf{X}_u^{\boldsymbol{\alpha}}(s), \boldsymbol{\alpha}) ds + \int_0^t \sigma_s dW_s^{\mathrm{u}},$$

$$\mathbf{X}_{\mathrm{u}}^{\boldsymbol{\mu},\boldsymbol{\beta}}(t) = \mathbf{X}_{\mathrm{u}}^{\boldsymbol{\mu},\boldsymbol{\beta}}(0) + \int_0^t \langle \mathbb{W}_{\boldsymbol{\beta}}[\mu_{\mathrm{v},s}], \boldsymbol{\psi} \rangle (\mathbf{X}_{\mathrm{u}}^{\boldsymbol{\beta}}(s)) ds + \int_0^t \boldsymbol{b}(s, \mathbf{X}_u^{\boldsymbol{\beta}}(s), \boldsymbol{\beta}) ds + \int_0^t \sigma_s dW_s^{\mathrm{u}}.$$

Given the dynamics above, the property of measure projection $\Psi$ induces the upper bound of generalized Wasserstein distance as follows:

$$\mathcal{W}_{t,\mathcal{M}}(\Phi(\boldsymbol{\mu}^{\boldsymbol{\beta}}), \Phi(\boldsymbol{\nu}^{\boldsymbol{\alpha}}))^2 \leq \mathbb{E}\left[\sup_{s \leq t} \|\mathbf{X}_u^{\boldsymbol{\mu},\boldsymbol{\beta}}(s) - \mathbf{X}_u^{\boldsymbol{\nu},\boldsymbol{\alpha}}(s)\|^2\right]$$

$$\leq b_0 \mathbb{E}\left[\int_0^t \int_{\mathcal{O}} \|\int_{\mathbb{R}^d} \boldsymbol{\psi}(\mathbf{X}_{\mathrm{u}}^{\boldsymbol{\mu},\boldsymbol{\beta}}(s), \mathbf{Y}) W_{\boldsymbol{\beta}}(\mathrm{u},\mathrm{v}) d\mu_{\mathrm{v},s}(\mathbf{Y})\right.$$

$$\left. - \int_{\mathbb{R}^d} \boldsymbol{\psi}(\mathbf{X}_{\mathrm{u}}^{\boldsymbol{\nu},\boldsymbol{\alpha}}(s), \hat{\mathbf{Y}}) W_{\boldsymbol{\alpha}}(\mathrm{u},\mathrm{v}) d\nu_{\mathrm{v},s}(\hat{\mathbf{Y}})\|^2 d\mathrm{v}_{\mathrm{Unif}} ds\right]$$

$$+ b_0 \mathbb{E}\left[\int_0^t \|\boldsymbol{b}(s, \mathbf{X}_u^{\boldsymbol{\mu},\boldsymbol{\alpha}}(s), \boldsymbol{\alpha}) - \boldsymbol{b}(s, \mathbf{X}_u^{\boldsymbol{\nu},\boldsymbol{\beta}}(t), \boldsymbol{\beta})\|^2 ds\right]$$

$$\leq 3b_0 (\mathrm{I} + \mathrm{II} + \mathrm{III}) + b_0 \mathrm{IV}, \tag{55}$$

where the first and second inequalities are induced from Holder's inequality and the Burkholder-Davis-Gundy (Chaintron & Diez, 2022) with some constant $b_0 > 0$. Following the assumptions in Section 8.2 and the modeling of graphons in Section 2, the first term (*i.e.*, I) can be upper-bounded in

the following estimation.

$$
\mathrm{I} := \mathbb{E}\left[\int_0^t \int_{\mathcal{O}} \left\|\left| \int_{\mathbb{R}^d} \left[ \boldsymbol{\psi}(\mathbf{X}_{\mathrm{u}}^{\boldsymbol{\nu},\boldsymbol{\alpha}}(s), \hat{\mathbf{Y}}) - \boldsymbol{\psi}(\mathbf{X}_{\mathrm{u}}^{\boldsymbol{\mu},\boldsymbol{\beta}}(s), \hat{\mathbf{Y}}) \right] W_{\boldsymbol{\alpha}}(\mathrm{u},\mathrm{v}) d\nu_{\mathrm{v},s}(\hat{\mathbf{Y}}) \right\|\right|^2 d\mathrm{v}_{\mathrm{Unif}} ds \right]
$$

$$
\leq \mathbf{Lip}(\boldsymbol{\psi})^2 \mathbb{E}\left[\int_0^t \int_{\mathcal{O}} W_{\boldsymbol{\alpha}}^2(\mathrm{u},\mathrm{v}) \int_{\mathbb{R}^d} \left|\left| \mathbf{X}_{\mathrm{u}}^{\boldsymbol{\nu},\boldsymbol{\alpha}}(s) - \mathbf{X}_{\mathrm{u}}^{\boldsymbol{\nu},\boldsymbol{\beta}}(s) \right|\right|^2 d\nu_{\mathrm{v},s}(\hat{\mathbf{Y}}) d\mathrm{v}_{\mathrm{Unif}} ds \right].
\tag{56}
$$

Given the fixed control $\boldsymbol{\alpha} = \bar{\boldsymbol{\alpha}}$, optimizing the last inequality requires estimating the (local) Lipschitz continuity of positional encoding $\boldsymbol{\psi}$:

$$
\mathbf{Lip}(\boldsymbol{\psi}(\cdot, \hat{\mathbf{Y}})) \leq \sup_{\mathrm{x} \in \mathbb{R}^d \setminus \{\hat{\mathbf{Y}}\}} \|\nabla \boldsymbol{\psi}(\mathrm{x}, \hat{\mathbf{Y}})\|
$$

$$
\leq H_{\boldsymbol{\psi}}(\bar{\boldsymbol{\alpha}}) \sup_{\mathrm{x} \in \mathbb{R}^d \setminus \{\hat{\mathbf{Y}}\}} \mathfrak{a}^{-2} \left\|\left( \mathbf{I}_d - \frac{2(\mathrm{x} - \hat{\mathbf{Y}}) \otimes_E (\mathrm{x} - \hat{\mathbf{Y}})}{\mathfrak{a}^2} \right) \right\|.
\tag{57}
$$

where $\mathfrak{a} = \|\mathrm{x} - \hat{\mathbf{Y}}\|$ and $\otimes_E$ denotes the Euclidean outer product. Following by the assumption (**H2**), Grönwall's inequality with the fact that $\mathbf{spec}(\nabla \boldsymbol{\psi}) := \lambda_1 \leq \max(1, -1)\mathfrak{a}^{-2}$, we have

$$
\mathrm{I} \leq C_1^2 H_{\boldsymbol{\psi}}^2(\bar{\boldsymbol{\alpha}}) \mathfrak{h}(\boldsymbol{\beta}) \mathbb{E}\left[\int_0^t \sup_{r \leq s} \left|\left| \mathbf{X}_{\mathrm{u}}^{\boldsymbol{\nu},\boldsymbol{\alpha}}(r) - \mathbf{X}_{\mathrm{u}}^{\boldsymbol{\nu},\boldsymbol{\beta}}(r) \right|\right|^2 ds \right].
\tag{58}
$$

Since each component $\boldsymbol{\psi}_i$ possesses the same spectral norm as $\boldsymbol{\psi}$, the second term can be upper-bounded with the improved definition of generalized Wasserstein distance in Eq (19):

$$
\mathrm{II} := \mathbb{E}\left[\int_0^t \int_{\mathcal{O}} \left\|\left| \int_{\mathbb{R}^d} \boldsymbol{\psi}(\mathbf{X}_{\mathrm{u}}^{\boldsymbol{\mu},\boldsymbol{\beta}}(s), \hat{\mathbf{Y}}) W_{\boldsymbol{\beta}}(\mathrm{u},\mathrm{v}) d[\nu_{\mathrm{v},s} - \mu_{\mathrm{v},s}](\hat{\mathbf{Y}}) \right\|\right|^2 d\mathrm{v}_{\mathrm{Unif}} ds \right]
$$

$$
\leq d|\mathcal{O}|C_1^2 \mathbb{E}\left[\sup_{\mathrm{u} \in \mathcal{O}} \max_{i \in \{1, \cdots, d\}} \int_0^t \left| \int_{\mathbb{R}^d} \frac{\boldsymbol{\psi}_i}{C_1}(\mathbf{X}_{\mathrm{u}}^{\boldsymbol{\mu},\boldsymbol{\beta}}(s), \cdot) W_{\boldsymbol{\beta}}(\mathrm{u},\mathrm{v}) d[\nu_{\mathrm{v},s} - \mu_{\mathrm{v},s}] \right|^2 ds \right]
\tag{59}
$$

$$
\leq d|\mathcal{O}|C_1^2 \mathfrak{h}(\boldsymbol{\beta}) \int_0^t \mathcal{W}_{s,\mathcal{M}}(\boldsymbol{\mu}^{\boldsymbol{\beta}}, \boldsymbol{\nu}^{\boldsymbol{\alpha}})^2 ds.
$$

Regarding the third term (*i.e.*, III), we have

$$
\mathrm{III} := \mathbb{E}\left[\int_0^t \int_{\mathcal{O}} \left|\left| \int_{\mathbb{R}^d} \boldsymbol{\psi}(\mathbf{X}_{\mathrm{u}}^{\boldsymbol{\mu},\boldsymbol{\beta}}(s), \hat{\mathbf{Y}}) |W_{\boldsymbol{\beta}} - W_{\boldsymbol{\alpha}}| d\nu_{\boldsymbol{v},s}(\hat{\mathbf{Y}}) \right|\right|^2 d\mathrm{v}_{\mathrm{Unif}} ds \right]
$$

$$
\leq (2C_1^2 \mathfrak{m}_2 H_{\boldsymbol{\psi}} + 1) \int_0^t \int_{\mathcal{O}^2} |W_{\boldsymbol{\beta}} - W_{\boldsymbol{\alpha}}|^2 d\mathrm{v}_{\mathrm{Unif}}^{\otimes 2}(\mathrm{u},\mathrm{v}) ds
\tag{60}
$$

$$
\leq (2C_1^2 \mathfrak{m}_2 H_{\boldsymbol{\psi}} + 1)|\mathbb{T}| d_{\mathfrak{g}}^2(W_{\boldsymbol{\beta}}, W_{\boldsymbol{\alpha}}).
$$

The upper-bound of last term can be directly obtained by the Lipschitz condition.

$$
\mathrm{IV} := \mathbb{E}\left[\int_0^t \|\boldsymbol{b}(s, \mathbf{X}_u^{\boldsymbol{\mu},\boldsymbol{\alpha}}(s), \boldsymbol{\alpha}) - \boldsymbol{b}(s, \mathbf{X}_u^{\boldsymbol{\nu},\boldsymbol{\beta}}(t), \boldsymbol{\beta})\|^2 ds \right]
$$

$$
\leq \mathbf{Lip}_{\boldsymbol{b}} \mathbb{E}\left[\int_0^t \sup_{r \leq s} \left|\left| \mathbf{X}_{\mathrm{u}}^{\boldsymbol{\mu},\boldsymbol{\alpha}}(r) - \mathbf{X}_{\mathrm{u}}^{\boldsymbol{\nu},\boldsymbol{\beta}}(r) \right|\right|^2 ds \right] + \mathbf{Lip}_{\boldsymbol{b}} \int_0^t \sup_{r \leq s} \|\boldsymbol{\alpha} - \boldsymbol{\beta}\|_{\nu_{\mathrm{u}}(s)}^2 ds.
\tag{61}
$$

By replacing each term with numerical constants $C_3, C_4, C_5$ in the aggregation of all four terms, we finally have the following upper-bounds related to $d_{\mathfrak{g}}$, $\mathcal{W}_{\mathcal{M}}$ and $L_2$-norm:

$$
\mathbb{E}\left[\sup_{s\leq t}\|\mathbf{X}_u^{\boldsymbol{\mu},\boldsymbol{\beta}}(s) - \mathbf{X}_u^{\boldsymbol{\nu},\boldsymbol{\alpha}}(s)\|^2\right] \leq 3b_0\,(\mathrm{I} + \mathrm{II} + \mathrm{III}) + b_0(\mathrm{IV})
$$

$$
\leq \underbrace{b_0(3C_1 H_{\boldsymbol{\psi}}(\bar{\boldsymbol{\alpha}})\mathfrak{h}(\boldsymbol{\beta}) + \mathbf{Lip}_{\boldsymbol{b}})}_{:=\log(C_3/t)}\mathbb{E}\left[\int_0^s \sup_{r\leq s}\left|\left|\mathbf{X}_u^{\boldsymbol{\nu},\boldsymbol{\alpha}}(r) - \mathbf{X}_u^{\boldsymbol{\nu},\boldsymbol{\beta}}(r)\right|\right|^2 dr\right]
$$

$$
+ \underbrace{(6b_0 C_1^2\mathfrak{m}_2 H_{\boldsymbol{\psi}} + 3b_0)|\mathbb{T}|}_{:=C_4}\,d_{\mathfrak{g}}^2(W_{\boldsymbol{\beta}}, W_{\boldsymbol{\alpha}})
$$

$$
+ \underbrace{\max(3b_0 d|\mathcal{O}|C_1\mathfrak{h}(\boldsymbol{\beta}), \mathbf{Lip}_{\boldsymbol{b}})}_{:=C_5}\left(\int_0^t \sup_{r\leq s}\|\boldsymbol{\alpha} - \boldsymbol{\beta}\|_{\nu_u(r)}^2 + \mathcal{W}_{s,\mathcal{M}}(\boldsymbol{\mu}^{\boldsymbol{\beta}}, \boldsymbol{\nu}^{\boldsymbol{\alpha}})^2 ds\right). \quad (62)
$$

Applying Grönwall's inequality to the above result in Eq (62) and the first inequality in Eq (55) shows that there exists a constant $C' = 3\max(C_3, C_4, C_5)$ such that

$$
\mathcal{W}_{t,\mathcal{M}}(\Phi(\boldsymbol{\mu}^{\boldsymbol{\beta}}), \Phi(\boldsymbol{\nu}^{\boldsymbol{\alpha}}))^2
$$

$$
\leq C'\left(d_{\mathfrak{g}}^2(W_{\boldsymbol{\beta}}, W_{\boldsymbol{\alpha}}) + \int_0^t \sup_{r\leq s}\|\boldsymbol{\alpha} - \boldsymbol{\beta}\|_{\nu_u(r)}^2 + \mathcal{W}_{s,\mathcal{M}}(\boldsymbol{\mu}^{\boldsymbol{\beta}}, \boldsymbol{\nu}^{\boldsymbol{\alpha}})^2 ds\right). \quad (63)
$$

Next, the aim is to show the upper-bound of $d_{\mathfrak{g}}^2$, $\|\boldsymbol{\alpha} - \boldsymbol{\beta}\|_{\boldsymbol{\nu}}^2$ and $\mathcal{W}_{\cdot,\mathcal{M}}$. To proceed, let us first examine the upper bounds of the cut norms for both exponential and cosinusoidal graphons as follows:

$$
\sup_{A,B}\left|\int_{A\times B} W_{\boldsymbol{\alpha}}(\mathrm{u},\mathrm{v})d\mathrm{u}d\mathrm{v}\right|^2 \leq \mathfrak{h}(\boldsymbol{\alpha}) = \begin{cases} W_0^2 + 2W_0(W_{1,l} + W_{2,l}) + (2/L)(\sum_l W_{1,l} + W_{2,l})^2 \\ (T/2)W_1^2\left(e^{-2T^{-1}|\mathbb{O}|} - 1\right). \end{cases}
$$

$$
(64)
$$

Modifying the upper-bound in Eq (64) by replacing $W_{\boldsymbol{\alpha}}$ with $\delta W := W_{\boldsymbol{\alpha}} - W_{\boldsymbol{\beta}}$, one can derive the following

$$
d_{\mathfrak{g}}^2(W_{\boldsymbol{\beta}}, W_{\boldsymbol{\alpha}}) = \|\delta W\|_{\mathfrak{g}}^2 \leq \max\left(11\mathbf{Lip}_W,\ (T/2)(e^{-2T^{-1}|\mathbb{O}|} - 1)\right)\|\boldsymbol{\alpha} - \boldsymbol{\beta}\|_{\boldsymbol{\nu}}^2. \quad (65)
$$

At each stage $\{m\}_{1\leq m\leq M}$ with the given sequence of probability measures $\{\boldsymbol{\nu}^{\boldsymbol{\alpha}_m}\}_{1\leq m\leq M}$, we substitute $\Phi(\boldsymbol{\mu}^{\boldsymbol{\beta}})$ and $\Phi(\boldsymbol{\nu}^{\boldsymbol{\alpha}})$ in Eq (63) with $\Phi\circ\Psi(\boldsymbol{\nu}^{\boldsymbol{\alpha}_{m+1}})$ and $\Phi\circ\Psi(\boldsymbol{\nu}^{\boldsymbol{\alpha}_m})$, respectively. Then, one can derive the following relation:

$$
\mathcal{W}_{t,\mathcal{M}}(\Phi\circ\Psi(\boldsymbol{\nu}^{\boldsymbol{\alpha}_m}), \Phi\circ\Psi(\boldsymbol{\nu}^{\boldsymbol{\alpha}_{m-1}}))^2 = \mathcal{W}_{t,\mathcal{M}}(\mathbf{Law}(\mathbf{X}^{\boldsymbol{\nu}_{\boldsymbol{\alpha}_{m+1}^*},\boldsymbol{\alpha}_{m+1}^*}), \mathbf{Law}(\mathbf{X}^{\boldsymbol{\nu}_{\boldsymbol{\alpha}_m^*},\boldsymbol{\alpha}_m^*}))^2
$$

$$
\leq C'\left(d_{\mathfrak{g}}^2(W_{\boldsymbol{\alpha}_{m+1}^*}, W_{\boldsymbol{\alpha}_m^*}) + \int_0^t \sup_{r\leq s}\|\boldsymbol{\alpha}_{m+1}^* - \boldsymbol{\alpha}_m^*\|_{\nu_u(r)} + \mathcal{W}_{s,\mathcal{M}}(\boldsymbol{\nu}^{\boldsymbol{\alpha}_{m+1}^*}, \boldsymbol{\nu}^{\boldsymbol{\alpha}_m^*})^2 ds\right).
$$

$$
\leq C'\left(\max\left(t + 11\mathbf{Lip}_W, t + (T/2)(e^{-2T^{-1}|\mathbb{O}|} - 1)\right)\right)\sup_t\|\alpha(t,\cdot,\theta^{m+1}) - \alpha(t,\cdot,\theta^m)\|_{\nu_u(t)}^2
$$

$$
+ C'\int_0^t \mathcal{W}_{s,\mathcal{M}}(\boldsymbol{\nu}^{\boldsymbol{\alpha}_{m+1}^*}, \boldsymbol{\nu}^{\boldsymbol{\alpha}_m^*})^2 ds
$$

$$
\leq \underbrace{C'\left(\max\left(t + 11\mathbf{Lip}_W, t + (T/2)(e^{-2T^{-1}|\mathbb{O}|} - 1)\right)\right)}_{:=C(t)\,\leq\,C(T)}\left(\sup_t \boldsymbol{r}_m\right)
$$

$$
+ C'\int_0^t \mathcal{W}_{s,\mathcal{M}}(\boldsymbol{\nu}^{\boldsymbol{\alpha}_{m+1}^*}, \boldsymbol{\nu}^{\boldsymbol{\alpha}_m^*})^2 ds,
$$

where the radius of metric ball (i.e., $\mathbf{r}_m := r_{u,t,m}$) was defined in the proof of Proposition 8.3. In the first equality, the controls $\boldsymbol{\alpha}$ are replaced with their optimal profiles $\boldsymbol{\alpha}^*$ following the definition of the operator $\Psi$ in. To set up the subsequent stage, we substitute a pair of controls $(\boldsymbol{\alpha}_{m+1}^*, \boldsymbol{\alpha}_m^*)$

with $(\boldsymbol{\alpha}_{m+1}, \boldsymbol{\alpha}_m)$ again. Next, we show the stability of the result obtained above for $M$-th stage by observing the upper bound of $M$-fold of the operator composition.

$$\mathcal{W}_{t,\mathcal{M}}([\Phi \circ \Psi]^{\circ M}(\boldsymbol{\nu}^{\boldsymbol{\alpha}_1}), [\Phi \circ \Psi]^{\circ M}(\boldsymbol{\nu}^{\boldsymbol{\alpha}_0}))^2$$

$$\leq C(t) \sup_t \mathbf{r}_m + C' \int_0^t \mathcal{W}_{s_0,\mathcal{M}}([\Phi \circ \Psi]^{\circ M-1}(\boldsymbol{\nu}^{\boldsymbol{\alpha}_1}), [\Phi \circ \Psi]^{\circ M-1}(\boldsymbol{\nu}^{\boldsymbol{\alpha}_0}))^2 ds^0 \Big)$$

$$\vdots \tag{66}$$

$$\leq \sum_{m=1}^M (C(t) \sup_t \mathbf{r}_m)^m + (C')^M \int_0^{s_0} \cdots \int_0^{s_M} \mathcal{W}_{s_m,\mathcal{M}}(\boldsymbol{\nu}^{\boldsymbol{\alpha}_{m+1}}, \boldsymbol{\nu}^{\boldsymbol{\alpha}_m})^2 d[\Pi^M](s^0, \cdots s^M).$$

where $d\Pi^m := ds^0 \otimes \cdots \otimes ds^m$ denotes $m$-product of Lebesgue measures $\{ds^m\}_{1 \leq m \leq M}$. Finally, we deduce that the supremum of the left-hand side can be controlled by

$$\lim_{M \to \infty} \sup_{t \in \mathbb{T}} \mathcal{W}_{t,\mathcal{M}}([\Phi \circ \Psi]^{\circ m}(\boldsymbol{\nu}^{\boldsymbol{\alpha}_1}), [\Phi \circ \Psi]^{\circ m}(\boldsymbol{\nu}^{\boldsymbol{\alpha}_0}))^2 \leq$$

$$+ \frac{C(T)^M (\sup_t \sup_m \mathbf{r}_m)^M - 1}{C(T)(\sup_t \sup_m \mathbf{r}_m) - 1} + \frac{(C'T)^M}{M!} \sup_{t \in \mathbb{T}} \mathcal{W}_{t,\mathcal{M}}(\boldsymbol{\nu}^{\boldsymbol{\alpha}_1}, \boldsymbol{\nu}^{\boldsymbol{\alpha}_0})^2 \longrightarrow 0. \tag{67}$$

where the learning rate $\gamma^m$ is chosen such that $\sup_t \sup_m \mathbf{r}_m \leq 1$ remains sufficiently small, and the last term in the inequality can be derived by modifying the following

$$\sup_{t \in \mathbb{T}} \mathcal{W}_{t,\mathcal{M}}(\Phi^{\circ m}(\nu^{\boldsymbol{\alpha}_1}), \Phi^{\circ m}(\nu^{\boldsymbol{\alpha}_0}))^2 \leq (C')^M \int_0^T \frac{(T-s)}{(m-1)!} \mathcal{W}_{s,\mathcal{M}}(\nu^{\boldsymbol{\alpha}_1}, \nu^{\boldsymbol{\alpha}_0})^2 ds. \tag{68}$$

The inequality in Eq (67) demonstrates that the sequence of operator compositions $\{[\Phi \circ \Psi]^{\circ m}\}_{m \leq M} : \mathcal{M} \to \mathcal{M}$ forms a Cauchy sequence, confirming the convergence of the proposed gradient system in the distributional sense. $\square$

## 8.6 Sampling Errors of Mean-field Predictors

Though not presented in the manuscript, the following result implies key theoretical conclusions: It demonstrates that the estimation errors for the neural agent, introduced by the sampled mean-field predictors (empirical measure) at the $m$-th gradient descent step, are kept within acceptable margins.

**Proposition 8.4.** *(Worst-case Estimation Error of neural agents) Let $\mathbb{Q}_n := \mathbb{Q}_n(\mathrm{u}, t) = (1/n) \sum_i \delta_{\mathbf{X}_{\mathrm{u}_i}^{\boldsymbol{\alpha}}(t)}$ and $\mathbb{Q} := \nu_{\mathrm{u}}(t)$ be an empirical law of mean-field predictors and its mean-field limit. Then, the worst-case approximation error can be upper bounded with probability $1 - \delta$ as follows:*

$$\sup_{\boldsymbol{\alpha}_m \in \mathbb{A}} \left\| \int \boldsymbol{\alpha}^m d(\mathbb{Q}_n - \mathbb{Q}) \right\|_E^2 \leq \sqrt{\frac{32T^3(1+\mathfrak{m}_2)^2}{n} \ln\left(\frac{1}{\delta}\right)}$$

$$+ 4 \left( \sqrt{\frac{32}{n}} 2^{(3d-2)/2} \left( \varepsilon \gamma^{m-1} \mathbf{Lip}_\alpha \|\nabla_\theta \mathbf{Y}_{\mathrm{u}}^{m-1,\boldsymbol{\alpha}_{m-1}}(t)\|_E \right)^{d/2} \frac{d+2}{4(d-2)} \right)^{(d/2+2)^{-1}}. \tag{69}$$

**Remark.** While the admissible control set $\mathbb{A}$ guarantees the diminishing behavior of $\|\nabla_\theta \mathbf{Y}_{\mathrm{u}}^{m-1,\boldsymbol{\alpha}_{m-1}}(t)\|_E$, the second term in Eq (69) approaches zero as $m$ becomes large, even when $n$ is small.

*Proof.* The proof follows the standard convergence analysis of empirical processes. Let us fix the temporal variable $t$ and the labels of mean-field predictors $\mathrm{u}$. Then, one can show that the supremum of Euclidean norm can be decomposed as follows:

$$\sum_j^d \sup_{\pi_j \circ \boldsymbol{\alpha} \in \mathbb{A}_m^j} |\mathbb{E}_{\mathbb{Q}_n} \pi_j \circ \boldsymbol{\alpha}_m - \mathbb{E}_{\mathbb{Q}} \pi_j \circ \boldsymbol{\alpha}_m| \leq \sum_j^d \sup_{g \in \mathbb{A}_m^j} |\mathbb{E}_{\mathbb{Q}_n} g - \mathbb{E}_{\mathbb{Q}} g| := \boldsymbol{\Gamma}_{\mathbb{A}_m^j}(\mathbb{Q}_{\mathfrak{N}}, \mathbb{Q}), \tag{70}$$

where $\mathbf{\Gamma}_{\mathbb{A}_m^j}$ denotes the *integral probability metric* (Müller, 1997) with respect to the set $\mathbb{A}_m^j$ which consists of $j$-th component of neural agents at $m$-th stage. Note that the the supremum in the second term is taken for all function $g$ lying in the set of parameterized function, *i.e.*, neural agent. Let us define $\mathfrak{p}, \mathfrak{q} : (\mathbb{R}^d)^n \to \mathbb{R}$ such that $(\mathbf{X}_{\mathrm{u}_1}^{\boldsymbol{\alpha}}(t), \cdots, \mathbf{X}_{\mathrm{u}_n}^{\boldsymbol{\alpha}}(t)) \overset{\mathfrak{p}}{\mapsto} \sup_g |(1/n) \sum_i g(\mathbf{X}_{\mathrm{u}_i}^{\boldsymbol{\alpha}}(t)) - \mathbb{E}_{\mathbb{Q}} g|$, and $(\mathbf{X}_{\mathrm{u}_1}^{\boldsymbol{\alpha}}(t), \cdots, \mathbf{X}_{\mathrm{u}_n}^{\boldsymbol{\alpha}}(t)) \overset{\mathfrak{q}}{\mapsto} \mathbb{E}_{\boldsymbol{\sigma}} \sup_g |(1/n) \sum_i \sigma_i g(\mathbf{X}_{\mathrm{u}_i}^{\boldsymbol{\alpha}}(t))|$ where $\{\sigma_i\}_{i \leq n}$ is a set of i.i.d Rademacher random variables. Then both $\mathfrak{p}$ and $\mathfrak{q}$ satisfies the following inequality:

$$\sup_t \max_{i \in \{1, \cdots n\}} \left| (\mathfrak{p}, \mathfrak{q})(\mathbf{X}_{\mathrm{u}_1}^{\boldsymbol{\alpha}}(t), \cdots, \mathbf{X}_{\mathrm{u}_{i-1}}^{\boldsymbol{\alpha}}(t), \mathrm{x}', \mathbf{X}_{\mathrm{u}_{i+1}}^{\boldsymbol{\alpha}}(t), \cdots \mathbf{X}_{\mathrm{u}_n}^{\boldsymbol{\alpha}}(t)) \right.$$
$$\left. - (\mathfrak{p}, \mathfrak{q})(\mathbf{X}_{\mathrm{u}_1}^{\boldsymbol{\alpha}}(t), \cdots, \mathbf{X}_{\mathrm{u}_n}^{\boldsymbol{\alpha}}(t)) \right| \leq \frac{4T \sup_{\mathrm{x}, t} \boldsymbol{\alpha}(t, \mathrm{x}; \theta)}{n}. \quad (71)$$

Following by the McDiarmid's inequality with respect to $\mathfrak{p}$, we have two concentration inequalities:

$$\exp\left( \frac{-n\varepsilon^2}{8T^2 \sup_{\mathrm{x}, t} \boldsymbol{\alpha}(t, \mathrm{x}; \theta)^2} \right) \geq \begin{cases} \mathbb{P}(\mathfrak{p} - \mathbb{E}\mathfrak{p} \geq \varepsilon) \\ \mathbb{P}(\mathfrak{q} - \mathbb{E}\mathfrak{q} \geq \varepsilon). \end{cases} \quad (72)$$

By applying the symmetrization inequality (Wellner et al., 2013), we have the following inequality with probability at least $1 - \delta$

$$\mathbf{\Gamma}_{\mathbb{A}_m^j}(\mathbb{Q}\mathfrak{N}, \mathbb{Q}) \leq \mathbb{E}\mathbf{\Gamma}_{\mathbb{A}_m^j}(\mathbb{Q}\mathfrak{N}, \mathbb{Q}) + \sqrt{\frac{8T^2 \sup_{\mathrm{x}, t} \boldsymbol{\alpha}(t, \mathrm{x}; \theta)^2}{n} \ln\left(\frac{1}{\delta}\right)}$$

$$\leq 2\tilde{\mathbb{E}}\mathbb{E}_{\boldsymbol{\sigma}} \left[ \sup_{g \in \mathbb{A}_m^j} \left| \frac{1}{n} \sum_i^n \sigma_i g(\mathbf{X}_{\mathrm{u}_i}^{\boldsymbol{\alpha}}(t)) \right| + \sqrt{\frac{8T^2 \sup_{\mathrm{x}, t} \boldsymbol{\alpha}(t, \mathrm{x}; \theta)^2}{n} \ln\left(\frac{1}{\delta}\right)} \right] \quad (73)$$

$$\leq 2 \underbrace{\mathbb{E}_{\boldsymbol{\sigma}} \left[ \sup_{g \in \mathbb{A}_m^j} \left| \frac{1}{n} \sum_i^n \sigma_i g(\mathbf{X}_{\mathrm{u}_i}^{\boldsymbol{\alpha}}(t)) \right| \right]}_{\mathcal{R}_m(\mathbb{A}_m^j, \{\mathbf{X}_{\mathrm{u}_n}^{\boldsymbol{\alpha}}(t)\})} + \sqrt{\frac{32T^3(1 + \mathfrak{m}_2)^2}{n} \ln\left(\frac{1}{\delta}\right)}$$

where the outer expectation is taken with respect to the randomness of mean-field predictors in the second line, and we apply McDiarmid's inequality in Eq (72) again to derive the last line. Following by the covering number of

$$\mathcal{R}_m(\mathbb{A}_m^j, \{\mathbf{X}_{\mathrm{u}_n}^{\boldsymbol{\alpha}}(t)\}) \leq \mathbb{E}_{\boldsymbol{\sigma}} \left[ \sup_{g \in \mathbb{A}_m^j} \left| \frac{1}{n} \sum_i^n \sigma_i g(\mathbf{X}_{\mathrm{u}_i}^{\boldsymbol{\alpha}}(t)) \right| \right]$$

$$\leq \inf_{\epsilon > 0} \left\{ 2\epsilon + \sqrt{\frac{32}{n}} \int_{\epsilon/4}^{\infty} \sqrt{\mathcal{H}(\tau, \mathbb{A}_m^j, \mathbb{L}_2(\mathbb{Q}_n))} \right\}$$

$$\leq \inf_{\epsilon > 0} \left\{ 2\epsilon + \sqrt{\frac{32}{n}} \int_{\epsilon/4}^{\infty} \left( \frac{2\mathbf{r}_m}{\tau} \right)^{d/2} d\tau \right\}$$

$$\leq \inf_{\epsilon > 0} \left\{ 2\epsilon + \sqrt{\frac{32}{n}} (2\mathbf{r}_m)^{d/2} (\epsilon/4)^{-d/2+1} (d/2 - 1)^{-1} \right\} \quad (74)$$

$$\leq \inf_{\epsilon > 0} \left\{ 2\epsilon + \sqrt{\frac{32}{n}} 2^{(3d-2)/2} \mathbf{r}_m^{d/2} \epsilon^{-d/2-1} (d - 2)^{-1} \right\}$$

$$= 4 \left( \sqrt{\frac{32}{n}} 2^{(3d-2)/2} \left( \varepsilon \gamma^{n-1} \mathbf{Lip}_\alpha \delta_\theta Y_{\mathrm{u}}^{m-1}(t) \right)^{d/2} \frac{d+2}{4(d-2)} \right)^{(d/2+2)^{-1}},$$

where we assume the data dimensionality is $d > 2$. The second line is a direct consequence of Theorem 16 (von Luxburg & Bousquet, 2004), the second inequality can be derived from the fact that $\mathbb{Q}_n$ is an empirical measure, and $\mathbb{A}_m^j$ is a metric ball of radius $\mathbf{r}_m$ embedded on finite-dimensional Hilbert space following by (**H4**). By setting $d = d'$, the last result comes from the definition of radius $\mathbf{r}_m$. $\qquad \square$

**Proposition 4.2.** *(Sampling Complexity) Let $\nu_t^N, \hat{\mu}_t$ probability measures defined in Eq (14). Then, there exist numerical constants $\mathfrak{c}, \mathfrak{c}_7, \mathfrak{c}_8, \mathfrak{c}_9 > 0, w > 0$ and $\kappa > 0$ such that the probability of squared 2-Wasserstein distance can be controlled as follows:*

$$\mathbb{P}\left[W_2^2(\nu_t^N, \hat{\mu}_t) \geq \epsilon\right] \leq \mathfrak{a}\left(\frac{1}{\epsilon^2}e^{-N\epsilon^2/4\mathfrak{c}} + \frac{1}{N}e^{-N\epsilon}\left(1 - \frac{128\omega\mathfrak{h}(\boldsymbol{\alpha})}{N}\right)^{-d/8} + \frac{1}{72^4\epsilon\sqrt{N}}\right), \quad (75)$$

$$\mathfrak{a} = \max\left(\mathfrak{c}_9, \frac{2\mathfrak{c}_7^{3/2}}{\kappa}\exp(\mathfrak{c}_4 e^{\frac{1}{2}\mathfrak{c}_1 T})\left(e^{\kappa T} - 1\right), \mathfrak{c}_9\exp(-4\mathfrak{c}_8)\right), \quad (76)$$

*where $u \in \mathcal{O}$, $t \in \mathbb{T}$ is arbitrary and $\mathfrak{h}(\boldsymbol{\alpha}) = \|W_{\boldsymbol{\alpha}}\|_{\mathfrak{g}}$ is a cut-norm[a] of the proposed graphons (i.e., exponential, cosinusoidal).*

---

[a]Eq. 64 clarifies the explicit upper-bound of the cut-norm for the proposed graphons.

**Remark.** The approach used in the proof to establish the concentration bound is largely inspired by the series of works on the measure concentration (Bolley et al., 2007; Budhiraja & Fan, 2017; Bayraktar & Wu, 2022; Bayraktar et al., 2023; Bayraktar & Wu, 2023), with slight modifications tailored to the structure of the proposed mean-field system. We intentionally omit some parts of the proofs in this work that have already been covered in the reference.

*Proof.* We divide the proof into separate steps.

**1. Estimation of Concentration Inequality.** For the controlled mean-field system via neural agents $\boldsymbol{\alpha}$, fix the the population $\boldsymbol{\nu}^{\boldsymbol{\alpha}}$ and its related control $\mathbf{X}_u^{\nu, \boldsymbol{\alpha}} = \mathbf{X}_u$ and let $u = i/n$ for the moment. First, let us define the following probability measures:

$$\nu_t^n := \frac{1}{n}\sum_i^n \delta_{\mathbf{X}_i^n(t)}, \quad \bar{\nu}_t^n := \frac{1}{n}\sum_i^n \delta_{\mathbf{X}_{(i/n)}(t)}, \quad \hat{\mu}_t = \int \nu_u(t)p(du), \quad \bar{\mu}_t^n = \frac{1}{n}\sum_i^n \nu_{u=i/n}(t). \quad (77)$$

Then, we analyze the law of difference between the following two mean-field dynamics:

$$\mathbf{X}_u(t) = \mathbf{X}_u(0) + \int_0^t \langle \mathbb{W}_{\boldsymbol{\alpha}}[\nu_{v,s}], \boldsymbol{\psi}\rangle(\mathbf{X}_u(s))ds + \int_0^t \boldsymbol{b}(s, \mathbf{X}_u(s), \boldsymbol{\alpha})ds + \int_0^t \sigma_s dW_s^u,$$

$$\mathbf{X}_i^n(t) = \mathbf{X}_{(i/n)}(0) + \int_0^t \langle \mathbb{W}_{\boldsymbol{\alpha}}[\delta_{v,s}], \boldsymbol{\psi}\rangle(\mathbf{X}_i^n(s))ds + \int_0^t \boldsymbol{b}(s, \mathbf{X}_i^n(s), \boldsymbol{\alpha})ds + \int_0^t \sigma_s dW_s^{(i/n)}.$$

Given that fact that the expectation of Ito's differential for mean-square error can be expressed as $d_{\mathbf{I}}\|A(t)\|^2 = 2\langle A(t), \boldsymbol{m}_A\rangle dt + 2\sigma A(t)dW_t + \sigma^2 dt$ where $\mathbb{R}^+ \ni \sigma$ and $\boldsymbol{m}_A$ are compensate and martingale part of $A(t)$, we get

$$d_{\mathbf{I}}\|\mathbf{X}_{(i/n)}(t) - \mathbf{X}_i^n(t)\|_E^2 = 2\delta\mathbf{X}(t)\cdot\left(\boldsymbol{b}(s, \mathbf{X}_i^n(s), \boldsymbol{\alpha}) - \boldsymbol{b}(s, \mathbf{X}_{i/n}(s), \boldsymbol{\alpha})\right)dt$$

$$\leq \left(\frac{1}{n}\sum_{j=1}^n W_{\boldsymbol{\alpha}}\left(\frac{i}{n}, \frac{j}{n}\right)\boldsymbol{\psi}_{\boldsymbol{\alpha}}(\mathbf{X}_i^n(t), \mathbf{X}_j^n(t)) - \hat{\mathbb{E}}\left[W_{\boldsymbol{\alpha}}\left(\frac{i}{n}, v\right)\boldsymbol{\psi}_{\boldsymbol{\alpha}}(\mathbf{X}_{(i/n)}(t), x)\right]\right) \quad (78)$$

$$\cdot 2\delta\mathbf{X}(t)dt$$

where we denote $\hat{\mathbb{E}} := \mathbb{E}_{v\sim p(v), x\sim \nu_{v=j/n}(t)}$ and $p(v) := w_\#[\mathbf{Unif}(\mathbb{O})]$, $\delta\mathbf{X}(t) := \mathbf{X}_{(i/n)}(t) - \mathbf{X}_i^n(t)$. Then, the dissipativity assumption gives

$$d_{\mathbf{I}}\|\delta\mathbf{X}(t)\|_E^2 \leq \mathrm{I} + \mathrm{II} + \mathrm{III} + \mathrm{IV} \quad (79)$$

For simplicity let us denote $W^{i,j} := W_{\boldsymbol{\alpha}}(i/n, i/j)$, and $W^{i,v} := W_{\boldsymbol{\alpha}}(i/n, v)$. Using the dissipativity of the proposed drift function. For the second first, one can get

$$\mathrm{I} := 2\delta\mathbf{X}(t)\cdot\left(\boldsymbol{b}(s, \mathbf{X}_i^n(s), \boldsymbol{\alpha}) - \boldsymbol{b}(s, \mathbf{X}_{i/n}(s), \boldsymbol{\alpha})\right) \leq -\mathfrak{c}_1\|\delta\mathbf{X}(t)\|_E^2 \quad (80)$$

By adding and subtracting new terms, we have

$$\text{II} := \left( \frac{1}{n} \sum_{j}^{n} W^{i,j} \left[ \boldsymbol{\psi}_{\boldsymbol{\alpha}}(\mathbf{X}_i^n, \mathbf{X}_j^n) - \boldsymbol{\psi}_{\boldsymbol{\alpha}}(\mathbf{X}_{(i/n)} - \mathbf{X}_{(j/n)}) \right] \right) \cdot \delta\mathbf{X}(t) \tag{81}$$

$$\leq \frac{\mathbf{Lip}_{\boldsymbol{b}}}{n} \sum_{j}^{n} |\delta\mathbf{X}(t)| \left( |\delta\mathbf{X}(t)| + |\mathbf{X}_j^n(t) - \mathbf{X}_{(j/n)(t)}| \right)$$

Similarly, the second term can be upper-bounded as follows:

$$\text{III} := \left( \frac{1}{n} \sum_{j}^{n} W^{i,j} \left[ \boldsymbol{\psi}_{\boldsymbol{\alpha}}(\mathbf{X}_{(i/n)}, \mathbf{X}_{(j/n)}) - \mathbb{E}_{\nu_{j/n}(t)} \boldsymbol{\psi}_{\boldsymbol{\alpha}}(\mathbf{X}_i^n, \cdot) \right] \right) \cdot \delta\mathbf{X}(t) \tag{82}$$

$$\leq |\delta\mathbf{X}(t)| \cdot \|W^{i,j}\|_{\infty} \|\mathcal{F}_{\text{III}}^i\|_E^2.$$

By adding and subtracting the term $W_{i,j}\mathbb{E}\boldsymbol{\psi}_{\boldsymbol{\alpha}}(\mathbf{X}_{i/n}(t), \cdot)$, the fourth term can be improved as

$$\text{IV} := \left( \frac{1}{n} \sum_{j}^{n} \left[ W^{i,j} \int \boldsymbol{\psi}_{\boldsymbol{\alpha}}(\mathbf{X}_{i/n}(t), \cdot) d\nu_{i/n}(t) - \int W^{i,\text{v}} \boldsymbol{\psi}_{\boldsymbol{\alpha}}(\mathbf{X}_{i/n}(t), \cdot) d\nu_{\text{v}}(t) \right] \right) \cdot \delta\mathbf{X}(t)$$

$$\leq \frac{1}{n} \sum_{j}^{n} \|W^{i,j}\|_{\infty} \left( C_1 \mathcal{W}_2(\nu_{i/n}(t), \nu_{\text{v}}(t)) + n_2 d_{\mathfrak{g}}(W^{i,j}, W^{i,\text{v}}) \right)$$

$$\leq |\delta\mathbf{X}(t)| \cdot \|W^{i,j}\|_{\infty} \|\mathcal{F}_{\text{IV}}^i\|_E^2 \xrightarrow{n \to \infty} 0. \tag{83}$$

Note that the last inequality tends to zero for large enough $n$. Aggregating all the terms and using the fact that $g'(t) \leq ag(t) + b$ implies $g(t) \leq \int e^{-a(t-s)} b \, ds$ and $d/dt \|g(t)\|_E^2 \leq (1/2)g(t)^{-1/2}\dot{g}(t)$, where $g(t) := (1/n) \sum_i^n \|\delta\mathbf{X}(t)\|_E^2$ and $a = (2\mathbf{Lip}_{\boldsymbol{b}} - \mathfrak{c}_1)$, $b := b(\mathcal{F}_{\text{III}}^i, \mathcal{F}_{\text{IV}}^i)$, we have

$$\mathcal{W}_2^2(\nu_t^n, \bar{\nu}_t^n) \leq \frac{1}{n} \sum_{i}^{n} \|\delta\mathbf{X}(t)\|_E^2$$

$$\leq \int_0^t e^{-(4\mathbf{Lip}_{\boldsymbol{b}} - 2\mathfrak{c}_1)(t-s)} \left( \sup_{i',j'} \|W^{i',j'}\|_{\infty}^2 \frac{1}{n} \sum_{i}^{n} \left| \|\mathcal{F}_{\text{III}}^i\|_E^2 + \|\mathcal{F}_{\text{IV}}^i\|_E^2 \right|^2 \right) ds. \tag{84}$$

$$\leq \underbrace{\int_0^t e^{-(4\mathbf{Lip}_{\boldsymbol{b}} - 2\mathfrak{c}_1)(t-s)} \left( \sup_{i',j'} \|W^{i',j'}\|_{\infty}^2 \frac{1}{n} \sum_{i}^{n} \|\mathcal{F}_{\text{III}}^i\|_E^2 + \|\mathcal{F}_{\text{IV}}^i\|_E^2 \right) ds}_{:= \text{V} + \text{VI}}$$

where the first inequality follows from the estimation of Wasserstein distance for empirical measures, and the last inequality can be derived by applying AM-GM inequality.

$$\mathbb{P}\left[ W_2^2(\nu_t^n, \hat{\mu}_t) \geq \epsilon \right] \leq \mathbb{P}\left[ \underbrace{W_2^2(\bar{\nu}_t^n, \hat{\mu}_t)}_{:= \text{VII}} \geq \epsilon/2 \right] + \mathbb{P}[\text{V} \geq \epsilon/4] + \underbrace{\mathbb{P}[\text{VI} \geq \epsilon/4]}_{=0, \; n \gg N}, \tag{85}$$

where the last term vanishes for small enough $\epsilon$, with large $N$.

**2. Estimation of Exponential** $e^{\lambda_{\exp}\|\mathbf{X}_{\text{u}}(t)\|_E^2}$**.** In this step, we derive the upper bound of the exponential for the square norm of mean-field predictors. We first apply the Ito's lemma to $e^{\lambda_{\exp}\|\mathbf{X}_{\text{u}}(t)\|_E^2}$ for arbitrary scalar $\lambda_{\exp} > 0$ and observe that

$$d_{\mathbf{I}} e^{\lambda_{\exp}\|\mathbf{X}_{\text{u}}(t)\|_E^2} =$$

$$\lambda_{\exp} e^{\lambda_{\exp}\|\mathbf{X}_{\text{u}}(t)\|_E^2} \left( 2\mathbf{X}_{\text{u}} \cdot (\boldsymbol{b} + \boldsymbol{b}_W) dt + \sigma_t (d + 2\lambda_{\exp}\|\mathbf{X}_{\text{u}}(t)\|_E^2) dt + \sigma_t dB_{\text{u}} \right). \tag{86}$$

where gradient and Laplace of exponential can be calculated as $\nabla e^{\lambda_{\exp}\|\mathbf{X}_{\mathrm{u}}(t)\|_E^2} = 2\lambda_{\exp} e^{\lambda_{\exp}\|\mathbf{X}_{\mathrm{u}}(t)\|_E^2}$ and $\Delta e^{\lambda_{\exp}\|\mathbf{X}_{\mathrm{u}}(t)\|_E^2} = 2\lambda_{\exp} e^{\lambda_{\exp}\|\mathbf{X}_{\mathrm{u}}(t)\|_E^2}(d + 2\lambda_{\exp} e^{\lambda_{\exp}\|\mathbf{X}_{\mathrm{u}}(t)\|_E^2})$. Taking expectation on both sides with the dissipative condition, we can show that there exist constants $\mathfrak{c}_2 = 2\lambda_{\exp}(-\mathfrak{c}_1 + \sigma_t\lambda_{\exp})$, $\mathfrak{c}_3 = \lambda_{\exp}\sigma_t d$ that directly gives following two inequalities

$$d_{\mathbf{I}}\mathbb{E}[e^{\lambda_{\exp}\|\mathbf{X}_{\mathrm{u}}(t)\|_E^2}] \le \mathbb{E}\left[e^{\lambda_{\exp}\|\mathbf{X}_{\mathrm{u}}(t)\|_E^2}\left(\mathfrak{c}_2\|\mathbf{X}_{\mathrm{u}}(t)\|_E^2 + \mathfrak{c}_3\right)\right]dt + \mathbb{E}\left[\int M_s dt\right], \qquad (87)$$

$$\sup_{t \le T}\|\mathbf{X}_{\mathrm{u}}(t)\|_E^2 \le \sup_{t \le T}\|y_{\mathrm{u}}\|^2 + N_t + \mathfrak{c}_1\int_0^t \|\mathbf{X}_{\mathrm{u}}(s)\|_E^2 ds \le \mathfrak{c}_4 e^{\mathfrak{c}_1 T}. \qquad (88)$$

where the second inequality is a direct consequence of Grownall's inequality, and $M_t$ and $N_t$ denote some martingale. Applying Grownall's inequality again, we have the desired result.

$$d_{\mathbf{I}}\mathbb{E}[e^{\lambda_{\exp}\|\mathbf{X}_{\mathrm{u}}(t)\|_E^2}] \le (\mathfrak{c}_5 + \mathfrak{c}_6\mathbb{E}[e^{\lambda_{\exp}\|\mathbf{X}_{\mathrm{u}}(t)\|_E^2}])dt, \qquad (89)$$

$$\mathbb{E}[e^{\lambda_{\exp}\|\mathbf{X}_{\mathrm{u}}(t)\|_E^2}] \le (\exp(\lambda_{\exp}\|y_{\mathrm{u}}\|_E^2) + \mathfrak{c}_5)\exp(\mathfrak{c}_6 T) \le (\mathfrak{c}_7)^2. \qquad (90)$$

where we used inequality $e^a + e^b \le \exp\left(\max(a,b) + \ln(1 + \exp(-|a-b|))\right) = (\mathfrak{c}_7)^2$ such that $a = \lambda_{\exp}\|\mathbf{X}_{\mathrm{u}}(t)\|_E^2 + \mathfrak{c}_6 T$, $b = \ln\mathfrak{c}_5 + \mathfrak{c}_6 T$. Note that the upper-bound of the term $\exp(\lambda_{\exp}\|y_{\mathrm{u}}\|_E^2)$ at initial time $t = 0$ determines the exponential integrability of the right-hand side above.

**3. Estimation of Probability** $\mathbb{P}[V \ge \epsilon/4]$. By the exponential Markov inequality with some constant $\lambda > 0$, Jensen's inequality, we obtain

$$\mathbb{P}[V \ge \epsilon/4] := \mathbb{P}\left[\int_0^t e^{-(4\mathbf{Lip}_b - 2\mathfrak{c}_1)(t-s)}\left(\sup_{i',j'}\|W^{i',j'}\|_\infty^2 \frac{1}{n}\sum_i^n \|\mathcal{F}_{\mathrm{III}}^i\|_E^2\right)ds > \epsilon/4\right]$$

$$\le \frac{1}{n}\sum_i^n e^{-\lambda\epsilon/4}\mathbb{E}\left[\int_0^t e^{-(4\mathbf{Lip}_b - 2\mathfrak{c}_1)(t-s)}\right. \qquad (91)$$

$$\left. \cdot \exp\left(\lambda\mathfrak{h}(\boldsymbol{\alpha})\|\frac{1}{n}\sum_j^n \boldsymbol{\psi}_{\boldsymbol{\alpha}}(\mathbf{X}_{(i/n)}, \mathbf{X}_{(j/n)}) - \mathbb{E}_{\nu_{j/n}(t)}\boldsymbol{\psi}_{\boldsymbol{\alpha}}(\mathbf{X}_{(i/n)}, \cdot)\|_E^2\right)ds\right].$$

Note that $\|\boldsymbol{\psi}_{\boldsymbol{\alpha}}(\mathrm{x}, \mathrm{y})\|_E \le \mathbf{Lip}_{\boldsymbol{\psi}}(\|\mathrm{x}\|_E + \|\mathrm{y}\|_E)$ have linear growth for all $\mathrm{x}, \mathrm{y} \in \mathbb{R}^d$ by the assumptions.

$$\mathbb{E}\left[\exp\left(\lambda\mathfrak{h}(\boldsymbol{\alpha})\left\|\frac{1}{n}\sum_j^n \boldsymbol{\psi}_{\boldsymbol{\alpha}}(\mathbf{X}_{(i/n)}, \mathbf{X}_{(j/n)}) - \mathbb{E}_{\nu_{j/n}(t)}\boldsymbol{\psi}_{\boldsymbol{\alpha}}(\mathbf{X}_{(i/n)}, \cdot)\right\|_E^2\right)\right]$$

$$\le \mathbb{E}\left[\exp\left(\frac{2\lambda\mathfrak{h}(\boldsymbol{\alpha})\mathbf{Lip}_{\boldsymbol{\psi}}}{n}\|\mathbf{X}_{(i/n)}\|_E^2 + 2\lambda\mathfrak{h}(\boldsymbol{\alpha})\left\|\frac{1}{n}F_{\boldsymbol{\psi}}\right\|_E^2\right)\right] \qquad (92)$$

$$\le \left(2\mathbb{E}\left[\exp\left(\frac{4\lambda\mathfrak{h}(\boldsymbol{\alpha})\mathbf{Lip}_{\boldsymbol{\psi}}}{n}\|\mathbf{X}_{(i/n)}\|_E^2\right)\right]\right)^{1/2}\left(2\mathbb{E}\left[\exp\left(2\zeta\left\|\frac{1}{n}F_{\boldsymbol{\psi}}\right\|_E^2\right)\right]\right)^{1/2}$$

where the last inequality can be derived by applying exponential AM-GM inequality

$$\mathbb{E}\left[\exp\left(2\zeta\left\|\frac{1}{n}F_{\boldsymbol{\psi}}\right\|_E^2\right)\right] = \mathbb{E}\left[\exp\left(\left\|\frac{2\sqrt{\zeta}}{n}\mathbf{Z}\right\|_E \cdot \|F_{\boldsymbol{\psi}}\|_E\right)\right]$$

$$\le \mathbb{E}\left[\exp\left(\omega\left\|\frac{2\sqrt{\zeta}}{n}\mathbf{Z}\right\|_E^2 + \frac{1}{4\omega}\|F_{\boldsymbol{\psi}}\|_E^2\right)\right] \qquad (93)$$

$$\le \left(2\mathbb{E}\left[\exp\left(\frac{8\omega\zeta}{n^2}\|\mathbf{Z}\|_E^2\right)\right]\right)^{1/2}\left(2\mathbb{E}\left[\exp\left((10n) \cdot \mathbf{Lip}_{\boldsymbol{\psi}}\|F_{\boldsymbol{\psi}}\|_E^2\right)\right]\right)^{1/2}\exp(\mathfrak{c}_4 e^{\mathfrak{c}_1 T})$$

$$\le 2\mathfrak{c}_7\left(1 - \frac{16\omega\zeta}{n^2}\right)^{-\frac{d}{4}} \cdot \exp(\mathfrak{c}_4 e^{\mathfrak{c}_1 T}),$$

where $\mathbf{Z} \sim \mathcal{N}(0, \mathbf{I}_d)$ is a standard Gaussian random vector. The last inequality is a direct consequence of the property of the moment generation function. The second line can be deduced from the fact that the discretized predictors $\mathbf{X}_{(i/n)}$ and $\mathbf{X}_{(j/n)}$ are i.i.d with the selection of $\omega > 0$, $\lambda_{\exp}$ and $\zeta$ satisfying the following:

$$\frac{1}{4\omega}\|F_\psi\|_E^2 \le n \cdot \mathbf{Lip}_\psi \left(5\|\mathbf{X}_{(i/n)}\|_E^2 + \exp(\mathfrak{c}_4 e^{\mathfrak{c}_1 T})\right) \tag{94}$$

$$\lambda_{\exp} := \max\left(\frac{4\lambda\mathfrak{h}(\boldsymbol{\alpha})\mathbf{Lip}_\psi}{n}, (10n)\mathbf{Lip}_\psi\right). \tag{95}$$

$$\zeta := 2\lambda\mathfrak{h}(\boldsymbol{\alpha}) > 0 \tag{96}$$

By aggregating all the terms, we finally have

$$\mathbb{E}\left[\exp\left(\lambda\mathfrak{h}(\boldsymbol{\alpha})\big|\big|\frac{1}{n}\sum_j^n \boldsymbol{\psi}_{\boldsymbol{\alpha}}(\mathbf{X}_{(i/n)}, \mathbf{X}_{(j/n)}) - \mathbb{E}_{\nu_{j/n}(t)}\boldsymbol{\psi}_{\boldsymbol{\alpha}}(\mathbf{X}_{(i/n)}, \cdot)\big|\big|_E^2\right)\right]$$

$$\le 2\mathfrak{c}_7^{3/2}\left(1 - \frac{16\omega\zeta}{n^2}\right)^{-\frac{d}{8}} \cdot \exp(\mathfrak{c}_4 e^{\frac{1}{2}\mathfrak{c}_1 T}) \tag{97}$$

Thus, the probability of $V$ larger than threshold $\epsilon/4$ can be written as follows:

$$\mathbb{P}[\mathrm{V} \ge \epsilon/4] \le \frac{2}{\kappa n}e^{-n\epsilon}\mathfrak{c}_7^{3/2}\left(1 - \frac{16\omega\zeta}{n^2}\right)^{-\frac{d}{8}} \cdot \exp(\mathfrak{c}_4 e^{\frac{1}{2}\mathfrak{c}_1 T})\left(e^{\kappa T} - 1\right), \tag{98}$$

$$\kappa = -(4\mathbf{Lip}_b - 2\mathfrak{c}_1), \quad \lambda = 4n. \tag{99}$$

**4. Estimation of Probability** $\mathbb{P}[\mathrm{VII} \ge \epsilon/2]$. Now, it remains to establish the upper bound of the probability related to VII. We modify the standard estimation of concentration probabilities of empirical measures as outlined in Bolley (2010). By the triangle inequality, the probability can be decomposed as

$$\mathbb{P}\left[\mathrm{VII} \ge \frac{\epsilon}{2}\right] \le \mathbb{P}\left[\sup_{\substack{h\Delta \le t \le (h+1)\Delta \\ 0 \le h \le \bar{M}-1}} \mathcal{W}_2^2(\bar{\nu}_t^n, \bar{\nu}_{h\Delta}^n) \ge \frac{\epsilon}{6}\right] + \mathbb{P}\left[\sup_{0 \le h \le \bar{M}-1} \mathcal{W}_2^2(\bar{\nu}_{h\Delta}^n, \bar{\mu}_{h\Delta}^n) \ge \frac{\epsilon}{6}\right] \tag{100}$$

where the temporal interval can be also decomposed as $\mathbb{T} = [0, \Delta] \cup [\Delta, 2\Delta] \cup \cdots \cup [(M-1)\Delta, T] \subseteq \bigcup_{h=0}^{M-1}[h\Delta, (h+1)\Delta]$. The first term of the right-hand side above can be bounded as

$$\mathbb{P}\left[\sup_{h\Delta \le t \le (h+1)\Delta} \mathcal{W}_2^2(\bar{\nu}_{t_1}^n, \bar{\nu}_{t_2}^n) \ge \frac{\epsilon}{6}\right] \le \mathbb{P}\left[\frac{1}{n}\sup_{0 \le t_1 \le t_2 \le \mathfrak{t}}\|\mathbf{X}_{i/n}(t_1) - \mathbf{X}_{i/n}(t_2)\|_E^2 \ge \frac{\epsilon}{6}\right]$$

$$\le \exp\left(-n\sup_{\zeta > 0}\left(\epsilon\zeta - \log\mathbb{E}\exp\left(\zeta\sup_{0 \le t_1 \le t_2 \le \mathfrak{t}}\|\mathbf{X}_{i/n}(t_1) - \mathbf{X}_{i/n}(t_2)\|_E^2\right)\right)\right) \tag{101}$$

The first line is induced as any measures $\nu_{(\cdot)}^n$ are empirical, and the next line can be induced by using Chebyshev's exponential inequality and the independence of the mean-field predictor. Denoting $\delta\mathbf{X}_{(i/n)} := \sup_{0 \le t_1 \le t_2 \le \mathfrak{t}}\|\mathbf{X}_{i/n}(t_1) - \mathbf{X}_{i/n}(t_2)\|_E^2$ for any $t_1 \le t_2\mathbb{T}$, we can further improve the right hand side by showing

$$\mathbb{E}\exp\left(\zeta\delta\mathbf{X}_{(i/n)}\right) \le \exp(\zeta^2\mathfrak{c}_8)\exp(2\zeta\delta\mathbf{X}_{(i/n)}) \le \exp(\zeta^2\mathfrak{c}_8)\left(1 + \hat{C}\Delta\right), \tag{102}$$

where we used the fact that $ax \le a^2b + 2ax$ for all $a, b, x \ge 0$. In order to show the upper bound of the first term in the last inequality (102), we used the result (4.6) Bolley (2010) tailored to our case under the assumption made in Section 8.2 for fixed $\mathrm{u}$ and $\boldsymbol{\alpha}$. Combining results, we have

$$\mathbb{P}\left[\sup_{\substack{h\Delta \le t \le (h+1)\Delta \\ 0 \le h \le \bar{M}-1}} \mathcal{W}_2^2(\bar{\nu}_t^n, \bar{\nu}_{h\Delta}^n) \ge \frac{\epsilon}{6}\right] \le \bar{M}\exp\left(-n\sup_{\zeta > 0}\left(\epsilon\zeta - \zeta^2\mathfrak{c}_8 - \log(1 + \hat{C}\Delta)\right)\right)$$

$$\le \bar{M}\exp\left(-\frac{n\epsilon^2}{4\mathfrak{c}_8} - \log(1 + \hat{C}\Delta)\right) \le \frac{\mathfrak{c}_9}{\epsilon^2}\exp\left(-\frac{n\epsilon^2 + 1}{4\mathfrak{c}_8}\right), \quad \begin{cases}\Delta = \exp(4\mathfrak{c}_8^{-1})\hat{C}^{-1}, \\ \bar{M} \le \mathfrak{c}_9/\epsilon^2.\end{cases} \tag{103}$$

For the second term of the right-hand side in (100), we first apply Boole's inequality of events to have

$$
\mathbb{P}\left[\sup_{0 \leq h \leq \bar{M}-1} \mathcal{W}_2^2(\bar{\nu}_{h\Delta}^n, \hat{\mu}_{h\Delta}) \geq \frac{\epsilon^2}{36}\right] \leq \overbrace{\mathbb{P}\left[\sup_{0 \leq h \leq \bar{M}-1} \mathcal{W}_2^2(\bar{\mu}_{h\Delta}^n, \hat{\mu}_{h\Delta}) \geq \frac{\epsilon^2}{72}\right]}^{\to 0, n \gg N}
$$
$$
+ \mathbb{P}\left[\sup_{0 \leq h \leq \bar{M}-1} \mathcal{W}_2^2(\bar{\nu}_{h\Delta}^n, \bar{\mu}_{h\Delta}^n) \geq \frac{\epsilon^2}{72}\right]
$$
$$
\leq \frac{\bar{M}\epsilon}{(72)^4\sqrt{n}} \leq \frac{\mathfrak{c}_9}{(72)^4\epsilon\sqrt{n}}. \tag{104}
$$

The second inequality can be deduced by the result of Theorem 1.5 Bolley (2010) with $d \leq d' = 4, (0,1) \ni \hat{\delta} = 2, p = 2, q = 4$. Then, there exists a constant $n_0 > 0$ such that $n \geq n_0 \max\left(\epsilon^{-16}, \epsilon\right)$ for any $\epsilon > 0$ and

$$
\sup_{\substack{t \in \mathbb{T} \\ i \leq N}} \mathbb{P}\left[W_2^2(\delta_{\mathbf{X}_{(i/n)}(t)}), \nu_{(i/n)}(t) \geq \frac{\epsilon^2}{72}\right] \leq \frac{\epsilon}{(72)^4\sqrt{n}}. \tag{105}
$$

where the quantity in (106) can be derived by proceeding similarly as in Step 2.

$$
\sup_{\substack{t \in \mathbb{T} \\ i \leq N}} \mathbb{E}\left[\|\mathbf{X}_{(i/n)}(t)\|_E^4\right] \leq \infty \tag{106}
$$

The first term in the first inequality is direct consequence of following result:

$$
\mathbb{E}\left[\|\mathbf{X}_{(i/n)}(t) - \mathbf{X}_{(i/n)}(s)\|_E^2\right] \propto |t - s|^2. \tag{107}
$$

Combining all the results for the probability bounds of V, VII for deduce the upper bound in (85),

$$
\mathbb{P}\left[W_2^2(\nu_t^n, \hat{\mu}_t) \geq \epsilon\right] \leq \frac{\mathfrak{c}_9}{(72)^4\epsilon\sqrt{n}} + \frac{\mathfrak{c}_9}{\epsilon^2}\exp(-4\mathfrak{c}_8)\exp\left(-\frac{n\epsilon^2}{4\mathfrak{c}_8}\right)
$$
$$
+ \frac{2}{\kappa n}e^{-n\epsilon}\mathfrak{c}_7^{3/2}\left(1 - \frac{128\omega\mathfrak{h}(\boldsymbol{\alpha})}{n}\right)^{-\frac{d}{8}} \cdot \exp(\mathfrak{c}_4 e^{\frac{1}{2}\mathfrak{c}_1 T})\left(e^{\kappa T} - 1\right). \tag{108}
$$

By setting $\mathfrak{a}_0$ as follows, the proof is complete.

$$
\mathfrak{a}_0 = \max\left(\mathfrak{c}_9, \frac{2\mathfrak{c}_7^{3/2}}{\kappa}\exp(\mathfrak{c}_4 e^{\frac{1}{2}\mathfrak{c}_1 T})\left(e^{\kappa T} - 1\right), \mathfrak{c}_9 \exp(-4\mathfrak{c}_8)\right). \tag{109}
$$

$\square$

Figure 6: Parallel Computation in Sampling Mean-field Predictors

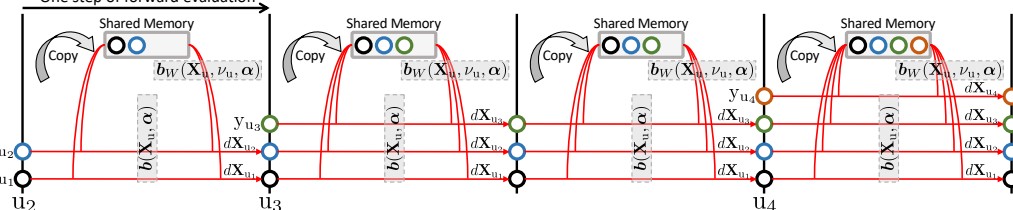

## 8.7 EXPERIMENTAL DETAILS

This section provides the implementation of mean-field predictors in detail.

**Experimental Setup.** Given that $\mathbb{T} := [0, T]$ is the time span of each continuous sequence data instance, our prediction task is to read the first (historical) $\alpha\%$ observations, $[0, (\alpha T/100)]$, and forecast the future $(1-\alpha)\%$ events, $[(\alpha T/100), T]$. We set $T$ to 100 for the MIT Humanoid Robot, 48 for MIMIC-II, and to 72 for the Beijing Air Quality dataset. The value of $\alpha$ is fixed at 80 across all datasets.

**Model Architecture.** In each forward step of $\mathbf{X}_u(t)$ from $t$ to $t + \Delta t$, a neural network takes $\mathbf{X}_u(t)$, $t$, and u as inputs and outputs $\boldsymbol{b}(\cdot, \boldsymbol{\alpha})$, $\boldsymbol{W}(\boldsymbol{\alpha})$, and $w$. In the first stage of the neural network, $\mathbf{X}_u(t)$ and $t$ are concatenated into a single vector, which is then projected into a hidden vector via a multilayer perceptron (MLP). This hidden vector is subsequently passed through a computation block consisting of several MLP layers with skip connections. Finally, after the computation block, the hidden vector is projected into $\boldsymbol{b}(\cdot, \boldsymbol{\alpha})$, $\boldsymbol{W}(\boldsymbol{\alpha})$, and $w$ using respective MLPs. In our architecture, each MLP is composed of two linear layers, with a Swish activation function positioned between them.

To process the labeling information u in the neural network, we apply adaptive normalization (Peebles & Xie, 2023). Specifically, instead of using fixed scale and shift parameters in the normalization layers of $\boldsymbol{\alpha}(.; \theta)$, we regress these parameters based on u. The adaptive normalization layers are placed between MLP layers. We find that this conditioning mechanism effectively incorporates the labeling information, outperforming the approach of simply concatenating u into input vectors.

After obtaining outputs from the neural networks, we evaluate $\boldsymbol{b}_W(\cdot, \boldsymbol{\alpha})$ for forward evaluation of SDEs. To derive $\boldsymbol{b}_W(\cdot, \boldsymbol{\alpha})$, we compute an exponential or cosine graphon $W$ using u and v where $v < t$. Next we calculate the projection $\mathbf{Proj}_{\mathcal{S}^{d-1}}(x - y) := (x - y)/\|x - y\|$ with $x = \mathbf{X}_u(t)$ and $y = \mathbf{X}_{v < t}(t)$. These values are then integrated into with $\boldsymbol{W}(\boldsymbol{\alpha})$ using Defintion 2.2 and Eq (2) or Eq (4) into $\boldsymbol{W}_{\boldsymbol{\alpha}}$ and $\boldsymbol{\psi}_{\boldsymbol{\alpha}}$, finally leading to $\boldsymbol{b}_W(\cdot, \boldsymbol{\alpha}) = \sum_{v < t} \boldsymbol{\psi}_{\boldsymbol{\alpha}}(\mathbf{X}_u(t), \mathbf{X}_v(t)) \boldsymbol{W}_{\boldsymbol{\alpha}}(u, v)$. After forward evaluation, we utilize $w$ to aggregate predictors by applying softmax. (*i.e.*, $\Lambda_t = \sum_{v < t} \texttt{Softmax}(w(u, v); \{w(t, u)\}_{u < t}) \mathbf{X}_u(t)$ where $\texttt{Softmax}(x \in S; S)$ represents the value of $x$ after applying the softmax operation to the entire set $S$ which includes x.)

**Parallel Computation.** Since the direct application of Alg. 1 is computationally intractable for large particle count $N$, we introduce novel parallel computing to efficiently sample proposed mean-field predictors, as described in Fig 6. At each step of forward evaluation, given all predictors $\mathbf{X}_u^{\alpha}$, each predictor can be processed independently using Eq (1). In other words, no predictor needs to wait for the others to complete their forward evaluation. By taking advantage of this property, at time $t$, we store all predictors with $u \leq t$ in the shared memory and forward predictors one step in parallel. This parallel implementation significantly decreases empirical computation time by reducing the number of iterations for forward evaluation from $\mathcal{O}(SN)$ to $\mathcal{O}(S)$ where $S$ is the number of steps for forward evaluation and $N$ is the number of sampled observations.

