# OpenReview forum: "Mean-field Continuous Sequence Predictors"
_ICLR.cc/2025/Conference — ICLR 2025 Conference Withdrawn Submission_

### Official Review · Reviewer_wyuC · 2024-10-15

**Soundness:** 3
**Presentation:** 2
**Contribution:** 3
**Rating:** 6
**Confidence:** 2

**Summary:**

The authors described a new method of modeling continuous time series through mean-field SDE where the mean-field interaction is computed over a graphon. It can be essentially be seen as an ensemble method (over a graphon) to model timeseries, where the graphon is semi-parametric with predetermined (temporal decay and cyclic) form. A stochastic controller is then applied to control the values of the parameters of the graphon. The authors also developed a gradient-based optimization algorithm to optimize the neural network used for stochastic control. Overall the paper has sound theoretical motivation and good empirical results.

**Strengths:**

1. Generally the idea is novel and nicely motivated.
2. Although it may seem incremental to add just the mean-field SDE based on graphon, the theoretical analysis and the related algorithms are non-trivial.
3. Demonstrated strong empirical improvement.

**Weaknesses:**

1. Confusion in notations. Throughout reading the manuscript, I would constantly be lost in new notations. Please make sure the notations are consistent.
2. Lack of Limitation as there is no discussion of the limitations of the proposed methods.
3. Lack of details in experiments and implementation.

**Questions:**

1. In definition 2.1, define $\psi$, such that in general the reader can know what it is, or maybe move definition 2.2? I was confused about \psi (although I can guess what it is), it will be easier for other readers to understand the equation if two definitions are written together.
2. Since the graphon is modeled by a neural network, is the linear assumption, where the mean-field drift and the Ito drift are linearly added, still necessary in Eq (1)? If not, is it possible to extend this framework to Mckean-Vlasov SDE to be more general than just mean field SDE ? (see[1], [2])
3. Notation discrepancy in Eq (3)? Eigen function was defined as $\phi_l$ but written as $\varphi_l$
4. Why do the authors assume the form of the graphon when $W_{\alpha} (u,v)$ are already modeled as neural networks? What is the implication of completely assuming the graphon to be a neural network without assuming its form? Can one still recover the temporal decay and cyclic properties when no such form is assumed?
5. Related to the above comment, [1] introduced implicit measure architecture that learns the mean field through a change of measure from the space of neural network weight to the observation space. Could this be implemented under this system without explicit assumption on the form of Graphon?
6. Is there a way to combine the temporal decay and the cyclic properties into one form of graphon?
7. Can the number of samples be incorporated in the cost function such that the most cost-effective number of samples can be obtained?

[1]: Yang, Haoming, et al. "Neural McKean-Vlasov Processes: Distributional Dependence in Diffusion Processes." International Conference on Artificial Intelligence and Statistics. PMLR, 2024.
[2]: Sharrock, Louis, et al. "Online parameter estimation for the McKean–Vlasov stochastic differential equation." Stochastic Processes and their Applications 162 (2023): 481-546.

---

### Official Review · Reviewer_kAGd · 2024-11-04

**Soundness:** 3
**Presentation:** 2
**Contribution:** 3
**Rating:** 6
**Confidence:** 2

**Summary:**

Authors cast the time-series prediction problem into a mean-field game, where they treat the time-series as arising from a controlled stochastic differential equation (mean-field graphon dynamic), which is given in terms of a continuum of mean-field predictors. Authors cast the problem thus to find an optimal control policy to the dynamic by solving the associated Bellman equation. The authors discuss how find such a policy by gradient descent in their mean-field game setting. Authors illustrate their method a number of real world data sets and perform two ablation studies.

**Strengths:**

- Seems to be the first paper to propose such a method to learn time series by casting as mean field game, which has been applied successfully in other areas of control and generative modeling/prediction.
- Paper relatively well organized (see weakness)
- Author provide numerical illustration on three datasets

**Weaknesses:**

- The way things are introduced is a bit confusing in Section 2. There is little motivation to the definition 2.1, which has limited reference to related literature. Many terms are not defined/explained until much later, e.g. the function $b$ is never explicitly defined or explained anywhere in the manuscript, and there are multiple overlapping uses of the variable $W$ with different meanings.
- Authors have specified little related literature to their work, e.g., connection of this work and (Liu et al. 2022) was not entirely clear to me.
- Empirical evaluation doesn’t include runtime results, e.g., the convergence rate of the solution to scheme presented in 3.2 (w.r.t. the training of other competing models is not specified)

Errata:
Line 92: Why is initial condition $y_u \sim p(u,y)$ measure dependent on $y$?

**Questions:**

- I am not very familiar with the mean field game literature, could the authors point to some references, and include earlier on in the paper, (before starting their own problem formulation)
- Could authors explain/illustrate how long it takes for their method to converge/train as compared to baselines.

---

### Official Review · Reviewer_Y1bs · 2024-11-04

**Soundness:** 3
**Presentation:** 2
**Contribution:** 3
**Rating:** 6
**Confidence:** 2

**Summary:**

The paper proposes a new model for predicting continuous sequences and discusses its theoretical underpinnings as well as its empirical evaluation on different benchmark datasets and in comparison to a range of baseline models.

**Strengths:**

- The proposed model seems theoretically well-motivated.
- The empirical evaluation shows a superior performance on different benchmarks compared to existing models.
- The paper describes two ablation studies that analyze the model's robustness to noise where it performs superior to the Mamba baseline and that analyze performance as the number of base predictor models increases, which leads to improved performance as predicted by the presented theory.

**Weaknesses:**

- This paper is very technical and builds on many rather sophisticated mathematical concepts that I would assume many readers not to be familiar with (and this in itself is of course not a weekness). I believe the presentation of the content could be improved so that the paper and the main concepts become more understandable, e.g. I believe a more high level introduction to the modelor some of the key concepts (e.g. graphons) would make it easier to follow the paper. Secondly, I sometimes was wondering about the notation of particular equations which were only clarified much later in the text. Examples for this are: $\mathcal{\nu}$, $\mathbb{W}$ or <> in definition 2.1, or the (subscript) E in $\mathbb{E} $$[||\mathbb{E}X_{u_\infty}^{α^*} (t) − y||^2_E]$ in the main text.

- I appreciate the overview figure 1 and can see that a lot of work went into that. However, I think the caption could be improved, e.g. there are three subfigures but the caption only mentions "left" and "right". Also what is the difference between"real observations" and the (observed?) values of u? And what is y in the legend?

**Questions:**

- What are some limitations of the method?
- How does the runtime compare to other baseline models that you compare to?

---

### Note · Authors · 2025-01-22

I have read and agree with the venue's withdrawal policy on behalf of myself and my co-authors.